**A 424-year tree-ring-based Palmer Drought Severity Index reconstruction of**

***Cedrus deodara* D. Don from the Hindu Kush range of Pakistan: Linkages to ocean**

**oscillations**

Sarir Ahmad[1, 2], Liangjun Zhu[1, 2], Sumaira Yasmeen[1, 2], Yuandong Zhang[3], Zongshan Li[4], Sami Ullah[5], Shijie Han[6, *], Xiaochun Wang[1, 2, *]

[1]Center for Ecological Research, Northeast Forestry University, Harbin 150040, China

[2]Key Laboratory of Sustainable Forest Ecosystem Management-Ministry of Education, School of Forestry, Northeast Forestry University, Harbin 150040, China

[3]Key Laboratory of Forest Ecology and Environment, State Forestry Administration, Institute of Forest Ecology, Environment and Protection, Chinese Academy of Forestry, Beijing 100091, China

[4]State Key Laboratory of Urban and Regional Ecology, Research Center for Eco-Environmental Sciences, Chinese Academy of Sciences, Beijing 100085, China

[5]Department of Forestry, Shaheed Benazir Bhutto University, Sheringal, Upper Dir 18000, Pakistan

[6]State Key Laboratory of Cotton Biology, School of Life Sciences, Henan University, Kaifeng 475001, China

**Corresponding authors**: Xiaochun Wang, E-mail: wangx@nefu.edu.cn and Shijie Han, E-mail: hansj@iae.ac.cn

**Abstract:** The rate of global warming has led to persistent drought. It is considered to be the preliminary factor affecting socioeconomic development under the background of dynamic forecasting of water supply and forest ecosystems in West Asia. However, long-term climate records in the semi-arid Hindu Kush range are seriously lacking. Therefore, we developed a new tree-ring width chronology of *Cedrus deodara* spanning the period of 1537–2017. We reconstructed the March–August Palmer Drought Severity Index (PDSI) for the past 424 y, going back to A.D. 1593. Our reconstruction featured nine dry and eight wet periods of 1593–1598, 1602–1608, 1631–1645, 1647–1660, 1756–1765, 1785–1800, 1870–1878, 1917–1923, and 1981–1995 and 1663–1675, 1687–1708, 1771–1773, 1806–1814, 1844–1852, 1932–1935, 1965–1969, and 1990–1999, respectively. This reconstruction was consistent with other dendroclimatic reconstructions in West Asia, thereby confirming its reliability. The multi-taper method and wavelet analysis revealed drought variability at periodicities of 2.1–2.4, 3.3, 6.0, 16.8, and 34.0–38.0 y. The drought patterns could be linked to the broad-scale atmospheric-oceanic variability, such as El Niño-Southern Oscillation, Atlantic Multidecadal Oscillation, and solar activity. In terms of current climate conditions, our findings have important implications for developing drought-resistant policies in communities on the fringes of the Hindu Kush mountain range in northern Pakistan.

Keywords: Tree ring, Climate change, Drought variability, El Niño-Southern Oscillation, Dendroclimatology, Atlantic Multidecadal Oscillation

## 1. Introduction

Numerous studies have shown that the intensity and frequency of drought events have increased owing to rapid climate warming (IPCC, 2013; Trenberth et al., 2014). Droughts have

serious adverse effects on social, natural, and economic systems (Ficklin et al., 2015; Yao and Chen, 2015; Tejedor et al., 2017; Yu et al., 2018). Globally, drought is considered to be the most destructive climate-related disaster, and it has caused billions of dollars in worldwide loss (van der Schrier et al., 2013; Lesk et al., 2016).

Pakistan has a semi-arid climate, and its agricultural economy is vulnerable to drought (Kazmi et al., 2015; Miyan, 2015). The South Asian summer monsoon (SASM) is an integral component of the global climate system (Cook et al., 2010). Owing to the annually recurring nature of the SASM, it is a significant source of moisture to the subcontinent and to surrounding areas such as northern Pakistan (Betzler et al., 2016). The active phase of the monsoon includes extreme precipitation in the form of floods and heavy snowfall, while the break phase mostly appears in the form of drought, thereby creating water scarcity. The active/break phases of the monsoon are also concurrent with El Niño-Southern Oscillation (ENSO) and land-sea thermal contrast (Xu et al., 2018; Sinha et al., 2007, 2011). The large-scale variability in sea surface temperature (SST) is induced in the form of Atlantic Multidecadal Oscillation (AMO), Pacific Decadal Oscillation (PDO), and some external forcing, i.e., volcanic eruption and greenhouse gases (Malik et al., 2017; Wei and Lohmann, 2012; Goodman et al., 2005).

The long-term drought from 1998 to 2002 reduced agricultural production, with the largest reduction in wheat, barley, and sorghum (from 60% to 80%) (Ahmad et al., 2004). Northern Pakistan is considered to contain the world's largest irrigation network (Treydte et al., 2006). The agricultural production and life of local residents are strongly dependent on monsoon precipitation associated with large-scale oceanic and atmospheric circulation systems, including ENSO, AMO, PDO, and others (Treydte et al., 2006; Cook et al., 2010; Miyan, 2015; Zhu et al., 2017). However, the current warming rate has changed the regional hydrological conditions,

thereby leading to an unsustainable water supply (Hellmann et al., 2016; Wang et al., 2017). It is not only critical for agricultural production, but also leads to forest mortality, vegetation loss (Martínez-Vilalta and Lloret, 2016), and increases the risk of wildfires (Abatzoglou and Williams, 2016). The degradation of grasslands and loss of livestock caused by drought affect the lifestyle of nomadic peoples, especially in high-altitude forested areas (Pepin et al., 2015; Shi et al., 2019).

The Hindu Kush Himalayan region (HKH) is the source of 10 major rivers in Asia, which provide water resources for 20% of the world's population (Rasul, 2014; Bajracharya et al., 2018). The region is particularly prone to droughts, floods, avalanches, and landslides, with more than 1 billion people being exposed to increasing frequency and serious risks of natural disasters (Immerzeel et al., 2010; Immerzeel et al., 2013). The extent of climate change in this area is significantly higher than the world average, which has seriously threatened the safety of life and property, traffic, and other infrastructure in the downstream and surrounding areas (Lutz et al., 2014). Dry conditions have been exacerbated by an increase in the frequency of heatwaves in recent decades (Immerzeel et al., 2010; IPCC, 2013). The trends in intensity and frequency of drought are very complex in the HKH, and there is no clear measuring tool to compute how long drought might persist. Climate uncertainty complicates the situation; for example, if the drought trend is increasing or decreasing (Chen et al., 2019). Most studies suggest that the wetting trend in the HKH will continue in the coming decades (Treydte et al., 2006). However, some extreme drought events in the region have been very serious and persistent (Gaire et al., 2017). Little has been done to examine the linkage between drought trends and large-scale oceanic climate drivers (Cook et al., 2003; Gaire et al., 2017). In northern Pakistan, instrumental climate records are inadequate in terms of quality and longevity (Treydte et al., 2006; Khan et al., 2019).

In high altitude, arid, and semi-arid areas, forest growth is more sensitive to climate change, so it is necessary to understand the past long-term drought regimes (Wang et al., 2008). Climate reconstruction is the best way to understand long-term climate change and expand the climate record to develop forest management strategies. Researchers have used multiple proxies, including ice cores, speleothems, lake sediments, historical documents, and tree rings, to reconstruct past short-term or long-term climate change. In addition, tree rings are widely used in long-term paleoclimatic reconstructions and future climate forecasting (Liu et al., 2004) because of their accurate dating, high resolution, wide distribution, easy access, long time series, and abundant environmental information (Esper et al., 2016; Zhang et al., 2015; Klippel et al., 2017; Shi et al., 2018; Chen et al., 2019)

Before 2000, there are few tree-ring studies in Pakistan. Bilham et al. (1983) found that tree rings of *Juniper* trees from the Sir Sar Range in the Karakoram have the potential to reconstruct past climate. Esper et al. (1995) developed a 1000-year tree-ring chronology at the timberline of Karakorum and found that temperature and rainfall are both controlling factors of *Juniper* growth. More Juniper tree-ring chronologies were developed at the upper timberline in the Karakorum (Esper, 2000; Esper et al., 2001; Esper et al., 2002). *Abies pindrow* and *Picea smithiana* were also used for dendroclimatic investigation in Pakistan (Ahmed et al., 2009; Ahmed et al, 2010). Recently, more studies on tree rings have been carried out in Pakistan (Ahmed et al., 2010; Ahmed et al., 2011; Khan et al., 2013; Akbar et al., 2014; Asad et al., 2017a; 2014; Asad et al., 2017b; Asad et al., 2018; Shad et al., 2019), but few have used tree rings to reconstruct the past climate, especially the drought index.

In this study, we collected drought-sensitive tree-ring cores of *Cedrus deodara* from the upper and lower HKH of Pakistan. These tree rings have good potential for dendroclimatic studies

(Yadav, 2013). Then, the March–August Palmer Drought Severity Index (PDSI) was reconstructed
for the past 424 y to examine the climatic variability and driving forces. To verify its reliability,
we compared our reconstructed PDSI with other available paleoclimatic records (Treydte et al.,
2006) near our research area. The intensity and drought mechanism in this area were also
discussed. This is the first time that the drought index has been reconstructed in northern Pakistan,
and the study can be used as a baseline for further tree-ring reconstruction in Pakistan.

**2. Material and Methods**
**2.1 Study area**
We conducted our research in the Hindu Kush (HK) mountain range of northern Pakistan
(35.36°N, 71.48°E; Fig. 1). Northern Pakistan has a subtropical monsoon climate. Summer is dry
and hot, spring is wet and warm, and in some high altitudes, it snows year-round. March is the
wettest month (with an average precipitation of 107.0 mm) while July or August is the driest
(with an average precipitation of 6.3 mm). July is the hottest month (mean monthly temperature
of 36.0 °C) and January is the coldest (mean monthly temperature of -0.8 °C) (Fig. 2).
The elevation of the study area ranges from 1070 to 7708 m, with an average elevation of
3500 m. The sampled Jigja site is located in the east slope of the mountain. The stand density is
relatively uniform with the dominant species. Among the tree species, *Cedrus deodara* is the
most abundant, with 156 individuals $hm^{-2}$ and basal area of 27 $m^2$ $hm^{-2}$. The Chitral forest is
mainly composed of *C. deodara*, *Juglans regia*, *Juniperus excelsa*, *Quercus incana*, *Quercus*
*dilatata*, *Quercus baloot*, and *Pinus wallichiana*. *C. deodara* was selected for sampling because
of its high dendroclimatic value (Khan et al., 2013). The soil at our sampling sites was acidic,
with little variation within a stand of forest. Similarly, the soil water holding capacity ranged
from 47%±2.4% to 62%±4.6% while the soil moisture ranged from 28%±0.57% to 57%±0.49%
(Khan et al., 2010).

Climate data such as monthly precipitation and temperature (1965–2016) were obtained

from the meteorological station of Chitral in northern Pakistan. The PDSI was downloaded from
data sets of the nearest grid point (35.36°N, 71.48°E) through the Climatic Research Unit (CRU
TS.3.22; 0.5° latitude × 0.5° longitude). The most common reliable period spanning from 1965
to 2016 was used (http://climexp.knmi.nl/) for dendroclimatic studies (Harris et al., 2014;
Shekhar, 2015).

The AMO index was downloaded from the KNMI Climate Explorer

https://climexp.knmi.nl/data/iamo_hadsst.dat over the period 1890-2016 (Mann et al., 2009). The
reconstructed June–August SASM data were downloaded from the Monsoon Asia Drought Atlas
http://drought.memphis.edu/MADA/TimeSeriesDisplay.aspx over the period of 1300–2005.

## 2.2 Tree-ring collection and chronology development

Tree-ring cores were collected in the Chitral forest from *C. deodara* trees. To maintain the

maximum climatic signals contained in the tree rings, undisturbed open canopy trees were
selected. One core per tree at breast height (approximately 1.3 m above the ground) was sampled
using a 5.15 mm diameter increment borer (Haglöf Sweden, Långsele, Sweden). In addition,
several ring-width series were also downloaded from the International Tree-Ring Data Bank
from the Bumburet forest and Ziarat forest (https://www.ncdc.noaa.gov/paleo-search/) collected
in 2006 (Fig. 1).

All the tree-ring samples were first glued and then progressively mounted, dried, and

polished according to a set procedure (Fritts, 1976; Cook and Kairiukstis, 1990). The preceding
calendar year was assigned and properly cross-dated. False rings were identified using a skeleton
plot and cross-dated, as mentioned in Stokes and Smiley (1968).
The cores were measured using the semi-automatic Velmex measuring system (Velmex,
Inc., Bloomfield, NY, USA) with an accuracy of 0.001 mm. The COFECHA program was then
used to check the accuracy of the cross-dating and measurements (Holmes, 1983). All false
measurements were modified and the cores that did not match the master chronology were not
used to develop the tree-ring chronology. For quality checks, COFECHA 2002 was used
(Holmes, 1998). The synthesized tree-ring width chronology (Fig. 3) was built using the R
program (Zang and Biondi, 2015). To preserve climate signals and avoid noise, appropriate
detrending was introduced. Biological trends in tree growth associated with tree age were
conservatively detrended by fitting negative exponential curves or linear lines (Fritts, 1976). The
tree-ring chronology was truncated where the expressed population signal (EPS) was larger than
0.85, which is a generally accepted standard for more reliable and potential climate signal results
(Wigley et al., 1984; Cook and Kairiukstis, 1990). The mean correlation between trees (Rbar),
mean sensitivity (MS), and EPS were calculated to evaluate the quality of the chronology (Fritts,
1976). Higher MS and EPS values indicated a strong response to climate change (Cook and
Kairiukstis, 1990).

**2.3 Statistical analysis**
Correlation analysis was conducted between the tree-ring index (TRI) and monthly
temperature, precipitation, and PDSI (from the previous June to current September; collected
from the nearby stations or downloaded from the KNMI). Then, the PDSI was reconstructed
according to the relationship between the TRI and climate variables. To test the validity and
reliability of our model, reconstruction was checked by the split-period calibration/verification
methods subjected to different statistical parameters, including reduction of error (RE),
coefficient of efficiency (CE), Pearson correlation coefficient ($r$), $R$-square ($R^2$), product mean
test (PMT), sign test (ST), and Durban-Watson test (DWT) (Fritts, 1976). The RE and CE have a
theoretical range of $-\infty$ to $+1$, but the benchmark for determining skill is the calibration and
verification period mean. Therefore, RE > 0 and CE > 0 indicate reconstruction skill in excess of
climatology (Cook et al., 1999). The PMT is used to test the level of consistency between the
actual and estimated values considering the signs and magnitudes of departures from the
calibration average (Fritts, 1976). The ST expresses the coherence between reconstructed and
instrumental climate data by calculating the number of coherence and incoherence value, which
was often used in previous studies (Fritts, 1976; Cook et al., 2010). The DWT is used to
calculate first-order autocorrelation or linear trends in regression residuals (Wiles et al., 2015).
RE and CE values larger than zero are considered skills (Fritts, 1976; Cook et al., 1999).

According to Chen et al. (2019), we defined the wet or dry years of our reconstruction with

a PDSI value greater than or less than the mean ± 1 standard deviation. The mean ± 1 standard
deviation is an easy method to calculate the dry and wet years, which has been observed in
different tree-ring PDSI reconstructions (Wang et al., 2008; Chen et al., 2019). We assessed the
dry and wet periods for many years based on strength and intensity.

Although there were few reconstructions in our study area, we compared our reconstruction

with other available drought reconstructions near the study area (Treydte et al., 2006). The multi-
taper method (MTM) was used for spectral analysis, and wavelet analysis was used to determine
the statistical significance of band-limited signals embedded in red noise by providing very high-
resolution spectral estimates that eventually provided the best possible option against leakage. To
identify the local climate change cycle, the background spectrum was used (Mann and Lees,

1996).


**3. Results**
**3.1 Main climate limiting factors for *Cedrus deodar***

The statistical parameters of the tree-ring chronologies, including MS (0.16), Rbar (0.59),

and EPS (0.94), indicated that there were enough common signals in our sampled cores and that
our chronology was suitable for dendroclimatic studies. According to the threshold of EPS (EPS
$> 0.85$), 1593–2016 was selected as the reconstruction period to truncate the period of 1537–
1593 of the chronology (Fig. 3).

The TRI was significantly positively correlated with the monthly PDSI ($p < 0.01$) (Fig. 4a).

However, the TRI was positively correlated with the precipitation in October of the previous
year and February–May of the current year and negatively correlated with the precipitation in
September of the previous year ($p < 0.001$). The TRI was significantly positively correlated ($p <$
0.001) with the minimum temperature in September and December of the previous year and
January and February of the current year (Fig. 4b). Similarly, the TRI was significantly
negatively correlated ($p < 0.001$) with the maximum temperature in January, October, and
December of the previous year and February–June of the current year. The TRI was only
significantly positively correlated with the maximum temperature in September (Fig. 4b).

**3.2 Reconstruction of past drought variation in northern Pakistan**

The correlation between the PDSI and the TRI was the highest from March to August,

thereby indicating that the growth of *C. deodara* was most strongly affected by drought before

and during the growing season. Based on the above correlation analysis results, the March–

August PDSI was the most suitable for seasonal reconstruction. The linear regression model

between the TRI and mean March–August PDSI for the calibration period from 1960 to 2016

was significant ($F = 52.4$; $p < 0.001$; adjusted $R^2 = 0.49$; $r = 0.70$). The regression model was as

follows:

$$Y = 5.1879x - 5.676$$

where $Y$ is the mean March–August PDSI and $x$ is the TRI.

The split calibration-verification test showed that the explained variance was higher during

the two calibration periods (1960–1988 and 1989–2016). For the calibration period of 1960–

2016, the reconstruction accounted for 39.2% of the self-calibrating Palmer Drought Severity

Index (SC-PDSI) variation (37.6% after accounting for the loss of degrees of freedom). The

statistics of $R$, $R^2$, ST, and PMT were all significant at $p < 0.05$, thereby indicating that the model

was reliable (Table 1). Here, the most rigorous RE and CE tests in the verification period were

all positive. Thus, these results made the model clearer and more robust in the PDSI

reconstruction.

The instrumental and reconstructed PDSIs of the HK mountains had similar trends and

parallel calibrations during short-term and long-term scales in the 20th century (Fig. 5).

However, the reconstructed scPDSI did not fully capture the magnitude of extremely dry or wet

conditions.

**3.3 Drought regime in the Hindu Kush mountain range, northern Pakistan for the past 424**

**years**

The dry periods were recorded as 1593–1598, 1602–1608, 1631–1645, 1647–1660, 1756–
1765, 1785–1800, 1870–1878, 1917–1923, and 1981–1995. Similarly, the wet periods were
recorded as 1663–1675, 1687–1708, 1771–1173, 1806–1814, 1844–1852, 1932–1935, 1965–
1969, and 1996–2003 (Fig. 5).
To verify the accuracy and reliability of our reconstruction, we compared our results with
the nearby precipitation reconstruction of Treydte et al. (2006) (Fig. 6). In the reconstruction of
Treydte et al. (2006), the high and low raw $\delta^{18}O$ values represent dry and wet conditions,
respectively, which is opposite to the PDSI. In most periods, our PDSI reconstruction and the
precipitation reconstruction of Treydte et al. (2006) showed good consistency (Fig. 6). However,
in some periods, they also showed inconsistent or even opposite changes in drought
reconstruction. For example, in 1865–1900, the reconstruction of Treydte et al. (2006) was very
wet, while our reconstruction was normal. In the periods of 1800–1810 and 1694–1702, the
reconstruction of Treydte et al. (2006) was very dry, but our reconstruction was wet (Fig. 6).
Spectral analysis of the historical PDSI changes in the HK mountains showed several
significant changes (95% or 99% confidence level) with periods of 33.0–38.0 (99%), 16.8 (99%),
and 2.0–3.0 (99%) y corresponding to significant periodic peaks (Fig. 7).
The spatial correlation analysis between our reconstructed PDSI and the actual PDSI from
May to August showed that our drought reconstruction was a good regional representation (Fig.
8). This showed that our reconstruction was reliable and could reflect the drought situation in the
region. In addition, the PDSI of low-frequency (the 31-year moving average) reconstruction had
good consistency with the AMO ($r = 0.53$; $p < 0.001$; 1890–2001) and SASM ($r = 0.35$; $p <$
0.001; 1608–1990), which indicated that these are the potential factors affecting the drought
patterns in the region (Fig. 9).

**4. Discussion**

**4.1 Drought variation in the Hindu Kush range of Pakistan**

The growth-climate relationship revealed the positive and negative influences of precipitation and summer temperature on growth. It indicated that water availability (PDSI) is the main limiting factor affecting the growth of *C. deodara*. Singh et al. (2006) reported that the previous October precipitation limits the growth of *C. deodara*, while Ahmed et al. (2011) found no such effect. Except for last August, November, and the current September, maximum temperature had a negative impact on the growth of *C. deodara*, while the minimum temperature did not. These results suggest that moisture conditions in April–July are critical to the growth of *C. deodara* in the study area (Borgaonkar et al., 1996; Khan et al., 2013). Chitral does not receive monsoon rains, which is why it is difficult to understand how trees respond to different moisture trends.

Here, we developed a 467 y (1550–2017) tree-ring chronology of *C. deodara* and reconstructed the 424 y (1593–2016) drought variability of the HK range in northern Pakistan. The peak years (narrow rings), namely 2002, 2001, 2000, 1999, 1985, 1971, 1962, 1952, 1945, 1921, 1917, 1902, and 1892, were recorded in our tree-ring record. Narrow ring formation occurs when extreme drought stress reduces cell division (Fritts et al., 1976; Shi et al., 2014). Therefore, the narrow rings were also consistent with the extreme drought years. Among them, 2001, 1999, 1952, and 1921 were identified by previous studies (Esper et al., 2003; Ahmed et al., 2010; Zafar et al., 2010; Khan et al., 2013; He et al., 2018). Sigdel and Ikeda (2010) reported that droughts occurred in 1974, 1977, 1985, 1993, the winter of 2001, and the summers of 1977, 1982, 1991, and 1992. Our PDSI reconstruction fully captured the widespread drought in Pakistan,

Afghanistan, and Tajikistan during 1970–1971 (Yu et al., 2014). The above drought disrupted
daily life and led to food and water shortages and livestock losses in high-altitude areas (Yadav,
2011; Yadav and Bhutiyani, 2013; Yadav et al., 2017). This drought might have also been due to
the failure of Western Disturbance precipitation (Hoerling et al., 2003). Similarly, the 17 wettest
years were observed from wide rings in 2010, 2009, 2007, 1998, 1997, 1996, 1993, 1931, 1924,
1923, 1908, 1696, 1693, 1691, 1690, 1689, and 1688. The floods of July 2010 were also captured
by our reconstruction, which affected approximately 20% of Pakistan (20 million people)
(Yaqub et al., 2015). The wet years of 1997, 1996, 1993, 1696, 1693, 1691, 1690, 1689, and
1688 were in agreement with the results of Khan et al. (2019). Similarly, the wet years of 1923,
1924, 1988, 2007, 2009, and 2010 coincided with the reconstruction of Chen et al. (2019).

As shown in Fig. 5, the mean of our reconstructed PDSI was below zero. There were two

possible reasons for this phenomenon. First, tree growth is more sensitive to drying than to
wetting. As a result, more drought information is recorded in ring widths. This leads to a drier
(less than zero) PDSI reconstructed with tree rings. This phenomenon exists in many tree-ring
PDSI reconstructions (Hartl-Meier et al., 2017; Wang et al., 2008). Second, the period (1960–
2016) used to reconstruct the equation was relatively dry. This caused the mean of the
reconstruction equation to be lower than zero (dry), thereby resulting in lower values for the
whole reconstruction. Therefore, when applying the PDSI data reconstructed by tree rings, its
relative value is relatively reliable, and the absolute value data can only be used after adjustment.
The adjustment method of the absolute value needs to be further studied.

Our reconstruction also captured a range of changes in climate mentioned in other studies

(Ahmad et al., 2004; Yu et al., 2014; Chen et al., 2019; Gaire et al., 2019). Our reconstruction
featured nine dry and eight wet periods of 1593–1598, 1602–1608, 1631–1645, 1647–1660,

1756–1765, 1785–1800, 1870–1878, 1917–1923, and 1981–1995 and 1663–1675, 1687–1708,

1771–1773, 1806–1814, 1844–1852, 1932–1935, 1965–1969, and 1990–1999, respectively. The

dry periods of 1598–1612, 1638–1654, 1753–1761, 1777–1793, and 1960–1985 and the wet

periods of 1655–1672, 1681–1696, 1933–1959, and 1762–1776 coincided with that

reconstructed by Chen et al. (2019) in northern Tajikistan. The most serious drought in 1871,

1881, and 1931, and the short-term drought from 2000 to 2002 mentioned by Ahmed et al.

(2004) were also found to be very dry in our reconstruction.

The dry period of 1645–1631 was also reported in tree-ring-based drought variability of the

Silk Road (Yu et al., 2014). Three mega-drought events in Asian history (Yadav, 2013; Panthi et

al., 2017; Gaire et al., 2019), namely the Strange Parallels Drought (1756–1768), East India

Drought (1790–1796), and Late Victorian Great Drought (1876–1878), were clearly recorded in

our reconstructed PDSI. This could mean that widespread drought on the continent could be

linked to volcanic eruptions (Chen et al., 2019). The wet period of 1995–2016 was very

consistent with that of Yadav et al. (2017). These results suggested that the long-term continuous

wet periods in 31 y out of the past 576 y (1984–2014) might have increased the mass of glaciers

in the northwest Himalaya and Karakoram mountains (Cannon et al., 2014). Therefore, we

speculated that the size of the HK glacier and the mass of glaciers near our study areas will

continue to increase if the wet trend continues.

Our PDSI reconstruction and the precipitation reconstruction of Treydte et al. (2006)

showed a strong consistency (Fig. 6), which proved that our reconstruction was reliable. The

discrepancies in some periods might have been caused because the PDSI is affected by

temperature and may not be completely consistent with precipitation (Li et al., 2015). The

inconsistency between the reconstruction of ring widths and oxygen isotopes in some periods

might also have been due to the different responses of radial growth and isotopes to disturbance (McDowell et al., 2002).

To test the consistency of the drought period, we compared this reconstruction with other drought and precipitation based on tree-ring- reconstructions in central Eurasia and China, which were adjacent to our study area, but none of them are completely matched. The dry periods of our reconstruction are similar to some periods of the reconstruction by Sun and Liu (2019) in 1629–1645 and 1919–1933. However, we found that our drought periods are more consistent with the drought periods of May–June reconstruction in the south-central Tibetan Plateau (He et al., 2018), such as 1593–1598 (1580–1598), 1647–1660 (1650–1691), 1785–1800 (1782–1807), and 1870–1878 (1867–1982). This difference may be due to differences in geographical location, species, and reconstruction indices, among others (Gaire et al., 2019). In addition, the lack of consistency between different data sets or regions might have been due to the dominance of internal climate variability over the impact of natural exogenous forcing conditions on multidecadal timescales (Bothe et al., 2019).

## 4.2 Linkage of drought variation with ocean oscillations

The results of wavelet and MTM analysis indicated that the low and high-frequency periods of drought in northern Pakistan may have a teleconnection with both large and small-scale climate oscillations (Fig. 7). The high frequency of the drought cycle (2.1–3.3 y) may be related to ENSO (van Oldenborgh and Burgers, 2005). The ENSO index in different equator Pacific regions has a significant positive correlation with our reconstructed drought index with a lag of 8 months (Table 2 and Fig. 10), so it further indicated that the water availability in this area may be related to large-scale climate oscillations. There is a lag effect of ENSO on drought in the study

area, the lag time is about 4-11 months. The lags in the ENSO impact are very complex and different in different regions (Vicente-Serrano et al., 2011). Therefore, the decrease of drought in our study area may be linked to the enhancement of ENSO activity. However, Khan et al. (2014) showed that most of northern Pakistan is in the monsoon shadow zone, and the Asian monsoon showed an overall weak trend in recent decades (Wang and Ding, 2006; Ding et al., 2008). Previous studies (Wang et al., 2006; Palmer et al., 2015; Shi et al., 2018; Chen et al., 2019) have confirmed that ENSO is an important factor regulating the hydrological conditions related to the AMO. In the past, severe famine and drought occurred simultaneously with the warm phase of ENSO, and these events were related to the failure of the Indian summer monsoon (Shi et al., 2014).

The middle-frequency cycle (16 y) might have been related to the solar cycle, which was similar to the results of other studies in South Asia (Panthi et al., 2017; Shekhar et al., 2018; Chen et al., 2019). Solar activity may affect climate fluctuations in the HK range in northern Pakistan (Gaire et al., 2017). The low-frequency cycle (36–38 y) might have been caused by the AMO, which is the anomalies of SST in the North Atlantic Basin (Fig. 9). Previous studies have shown that the AMO may alter drought or precipitation patterns in North America (Mccabe et al., 2004; Nigam et al., 2011) and Europe (Vicente-Serrano and López-Moreno, 2008). Although our study area is far from the Atlantic Ocean, it may also be affected by the AMO (Lu et al., 2006; Wang et al., 2011; Yadav, 2013). Lu et al. (2006) found that the SST anomalies in the North Atlantic (such as the AMO) can affect the Asian summer monsoon. Goswami et al. (2006) reported the mechanism of the influence of the AMO on Indian monsoon precipitation. The warm AMO appears to cause late withdrawal of the Indian monsoon by strengthening the meridional gradient of the tropospheric temperature in autumn (Goswami et al., 2006; Lu et al.,

2006). Yadav (2013) suggested the role of the AMO in modulating winter droughts over the western Himalayas through the tropical Pacific Ocean. Wang et al. (2009) pointed out that the AMO heats the Eurasian middle and upper troposphere in all four seasons, thereby resulting in weakened Asian winter monsoons and enhanced summer monsoons. This is consistent with the findings that the AMO affects the climate in China, which is made possible by the Atlantic-Eurasia wave train from the North Atlantic and is increased owing to global warming (Qian et al., 2014). Further work is still needed to determine the connections between the Pacific and Atlantic oceans and how the two are coupled through the atmosphere and oceans to affect drought in Asia.

Dimri (2006) found that the precipitation surplus in winter from 1958 to 1997 was related to the significant heat loss in the northern Arabian Sea, which was mainly due to intensification of water vapor flow in the west and the enhancement of evaporation. As a result, large-scale changes in Atlantic temperature could also regulate the climate of western Asia. Our result was supported by other dendroclimatic studies (Sano et al., 2005; Chen et al., 2019). Precipitation in the Mediterranean, Black Sea (Giesche et al., 2019), and parts of northern Pakistan showed an upward trend from 1980 to 2010, but precipitation in the HK mountains range was received from the Indian winter monsoon (December–March) and the rain shadows in summer (Khan et al., 2013). Predicting different patterns of the climate cycle is difficult. In addition, the flow of the Upper Indus Basin (UIB) depends on changes in the ablation mass (Rashid et al., 2018; Rao et al., 2018); small changes in the ablation mass may eventually lead to changes in water quality and quantity. The UIB is considered a water tower in the plain (Immerzeel et al., 2010), so the HK mountains are particularly important for extending the proxy network to improve the understanding of different climatic behaviors.

The drought regimes in the HK range in northern Pakistan may be linked to regional, local,
and global climate change. We only studied the response of *C. deodara* to different changes in
climate in the Chitral region of northern Pakistan. Therefore, we suggest that further high-
resolution and well-dated records are needed to augment the dendroclimatic network in the
region.

**5. Conclusion**
Based on the significance of the tree-ring widths of *C. deodara*, we developed a 467 y
chronology (1550–2017). Considering that the EPS threshold was greater than 0.85 (> 13 trees),
we reconstructed the current March–August PDSI from 1593 to 2016. Our reconstruction
captured different drought changes at different time scales in the HK range, Pakistan. Three
historic mega-drought events, namely the Strange Parallels Drought (1756–1768), East India
Drought (1790–1796), and Late Victorian Great Drought (1876–1878), were captured by our
reconstructed PDSI. These large-scale and small-scale droughts might have been caused by cold
or hot climate. Our results are consistent with other dendroclimatic records, which further
supports the feasibility of our reconstruction. In addition, owing to the different climate change
patterns in the region, we suggest extending the different proxy networks to understand the
remote teleconnection across the continent on multidecadal to centennial timescales to meet
future climate challenges.

**Acknowledgments**
This research was supported by the Key Project of the China National Key Research and
Development Program (2016YFA0600800), Fundamental Research Funds for the Central
Universities (2572019CP15 and 2572017DG02), Open Grant for Eco-Meteorological Innovation
Laboratory in northeast China, China Meteorological Administration (stqx2018zd02), and
Chinese Scholarship Council. We appreciate the staff of the International Office of Northeast
Forestry University for their excellent services. We thank Dr. Muhmmad Usman, Mr. Shahid
Humayun Mirza, and Dr. Nasrullah Khan for their help in revising the manuscript. We also
appreciate Mr. Muhmmad Arif, Sher Bahder, Wali Ullah, and Mushtaq Ahmad for their help
with the fieldwork.

**Data availability**
The reconstructed PDSI can be obtained from the supplementary file of this paper. The tree-ring
width data used in this study can be download from the International Tree-Ring Data Bank.

**Author contributions**
Xiaochun Wang and Sarir Ahmad initiated this study. Sarir Ahmad and Sami Ullah collected
samples in the field. Liangjun Zhu and Sumaira Yasmeen cross-dated and measured the samples.
Sarir Ahmad and Xiaochun Wang wrote the manuscript. Liangjun Zhu, Yuandong Zhang,
Zongshan Li, and Shijie Han revised the manuscript.

**Competing interest**
The authors declare that they have no conflict of interest.

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

**Figure captions**

**Fig. 1** Map of the weather stations (Drosh station) and sampling sites in the Chitral, HinduKush Mountains, Pakistan. Different colors represent the elevation changes of the study area.

**Fig. 2** Monthly maximum, mean, minimum temperature ($^{\circ}$C) and total precipitation (mm) in the Drosh Weather Station (35.07° N, 71.78° E, 1465 m), Pakistan (1965-2013).

**Fig. 3** The regional tree-ring width chronology from 1550 to 2017 in the Chitral, HinduKush Mountains, Pakistan. The gray area represents the sample depth.

**Fig. 4** Pearson correlation coefficients between the tree-ring index of *C. deodara* and monthly total precipitation (1965-2013) and scPDSI (1960-2013) (a) and monthly maximum and minimum temperature (1965-2013) (b) from June of the previous year to September of the current year. Significant correlations ($p<0.05$) are denoted by asterisks. The "previous" and "current" represents the previous and current year, respectively.

**Fig. 5** The scPDSI reconstruction in the Chitral HinduKush Mountain, Pakistan. (a) Comparison between the reconstructed (black line) and actual (red line) scPDSI; (b) The variation of annual (black solid line) and 11-year moving average (red bold line) Mar-Aug scPDSI from 1593 to 2016 with mean vale $\pm$ one standard deviation (black dash lines).

**Fig. 6** Comparison of our PDSI reconstruction (a) with the precipitation reconstruction (tree-ring $\delta^{18}$O) of Treydte et al. (2006) (b) in northern Pakistan. Purple and brown shaded areas represent the consistent wet and dry periods in the two reconstructions, respectively. Two correlation coefficients ($r = -0.24$ and $r = -0.11$) are the correlation of two original annual resolution reconstruction series and two 11-year moving average series, respectively.

**Fig. 7** The Multi-taper method spectrums of the reconstructed scPDSI from 1593 to 2016. Red and green line represents the 95% and 99% confidence level, respectively. The figures

above the significant line represents the significant periods of drought at 95% confidence
level.
**Fig. 8** (a) Spatial correlation between the actual May-August PDSI and the reconstructed May-
August scPDSI (1901-2017). (b) The wavelet analysis of the reconstructed scPDSI in the
Chitral HinduKush Ranges, Pakistan. The 95% significance level against red noise was
shown as a black contour.
**Fig. 9** (a) Comparison of the 31-year moving average series between the reconstructed Mar-Aug
scPDSI and the AMO index during the common period (1890-2001, Mann et al., ); (b)
Comparison of the 31-year moving average series between the reconstructed Mar-Aug
scPDSI and the South Asian Summer Monsoon index from June to August (JJA-SASM)
(Cook et al., 2010) during the common period (1608-1990).
**Fig. 10** The field correlation between the monthly HadISST1 sea surface temperature and
reconstructed PDSI with a lag of 8 months calculated by the KNMI Climate Explorer (1870-
2016). The contours with $p > 0.05$ were masked out.




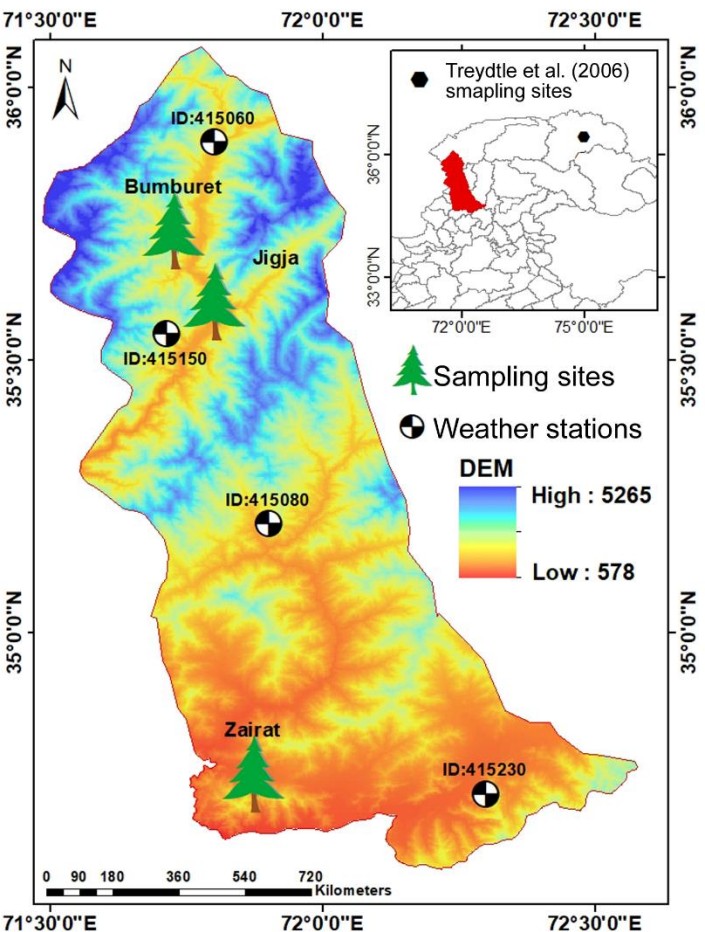


**Fig. 1** Map of the weather stations (Drosh station) and sampling sites in the Chitral, HinduKush
Mountains, Pakistan. Different colors represent the elevation changes of the study area.


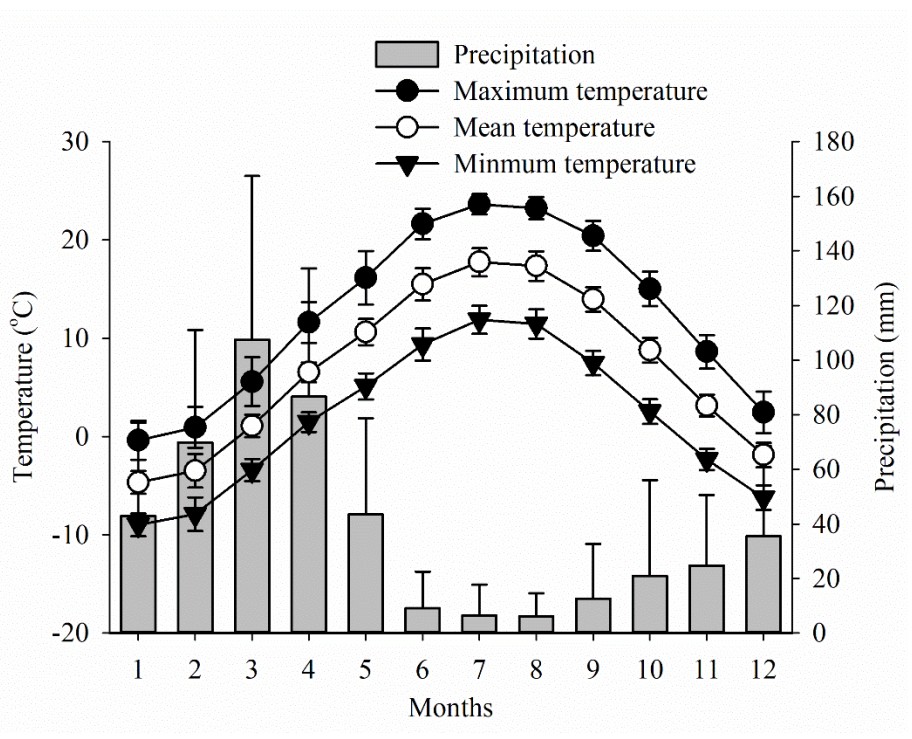


**Fig. 2** Monthly maximum, mean, minimum temperature (°C) and total precipitation (mm) in the
Drosh Weather Station (35.07° N, 71.78° E, 1465.0 m), Pakistan (1965-2013).

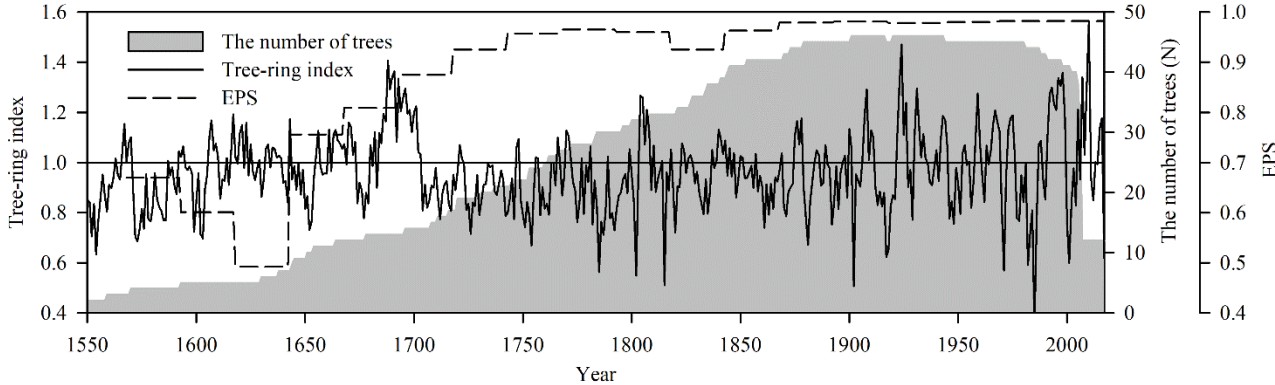


**Fig. 3** The regional tree-ring width chronology from 1550 to 2017 in the Chitral, HinduKush
Mountains, Pakistan. The gray area represents the sample depth.

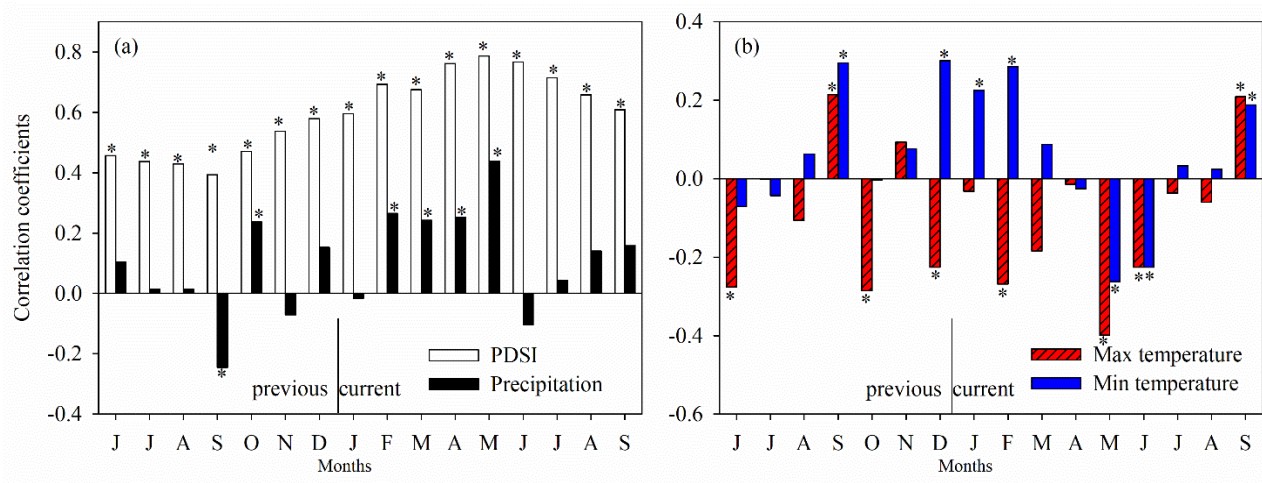

**Fig. 4** Pearson correlation coefficients between the tree-ring index of *C. deodara* and monthly

total precipitation (1965-2013) and scPDSI (1960-2013) (a) and monthly maximum and

minimum temperature (1965-2013) (b) from June of the previous year to September of the

current year. Significant correlations ($p<0.05$) are denoted by asterisks. The "previous" and

"current" represents the previous and current year, respectively. The data of monthly

precipitation, maximum temperature and minimum temperature were obtained from the

meteorological station of Chitral in northern Pakistan. The PDSI data was download from data

sets of the grid point (35.36 °N, 71.48 °E) through the Climatic Research Unit (CRU TS.3.22;

0.5° latitude × 0.5° longitude)

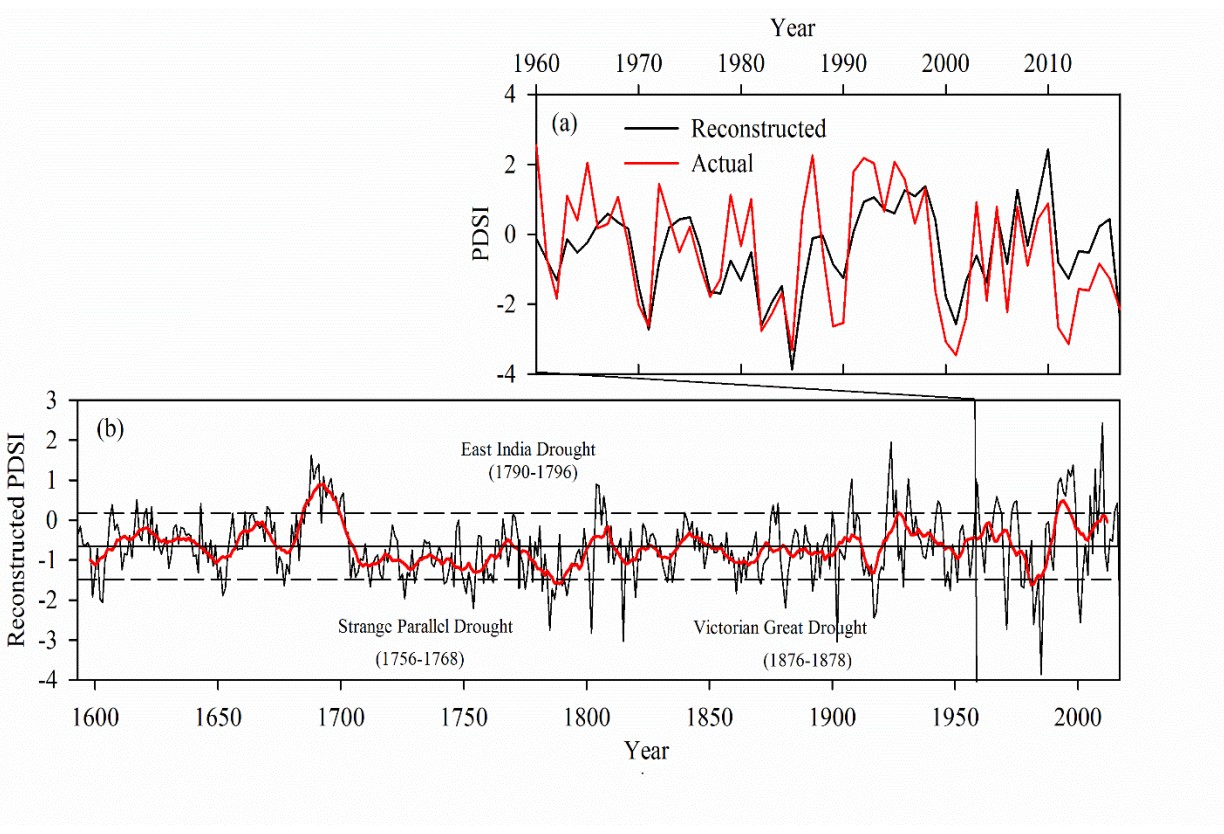

841

**Fig. 5** The scPDSI reconstruction in the Chitral HinduKush Mountain, Pakistan. (a) Comparison

between the reconstructed (black line) and actual (red line) scPDSI; (b) The variation of annual

(black solid line) and 11-year moving average (red bold line) Mar-Aug scPDSI from 1593 to

2016 with mean vale $\pm$ one standard deviation (black dash lines).


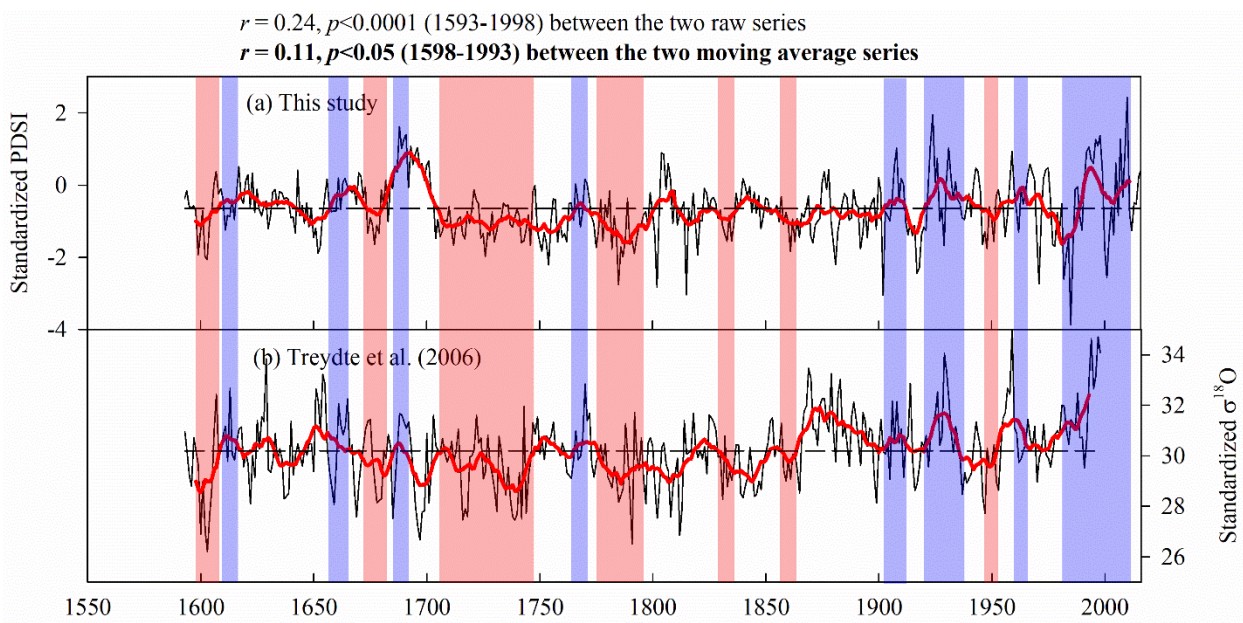


**Fig. 6** Comparison of our PDSI reconstruction (a) with the reversed precipitation reconstruction
(tree-ring $\delta^{18}O$) of Treydte et al. (2006) (b) in northern Pakistan. Purple and brown shaded areas
represent the consistent wet and dry periods in the two reconstructions, respectively. Two
correlation coefficients ($r = 0.24$ and $r = 0.11$) are the correlation of two original annual
resolution reconstruction series and two 11-year moving average series, respectively.

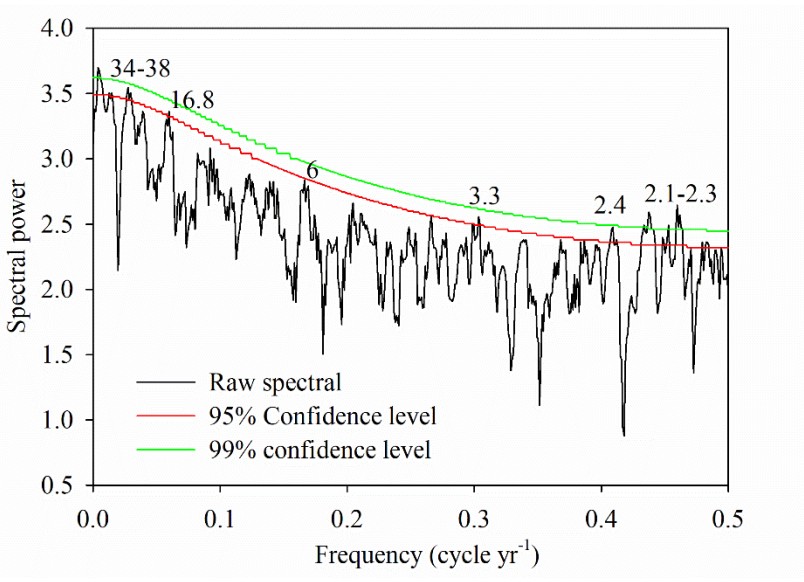


**Fig. 7** The Multi-taper method spectrums of the reconstructed scPDSI from 1593 to 2016. Red and
green line represents the 95% and 99% confidence level, respectively. The figures above the
significant line represents the significant periods of drought at 95% confidence level.

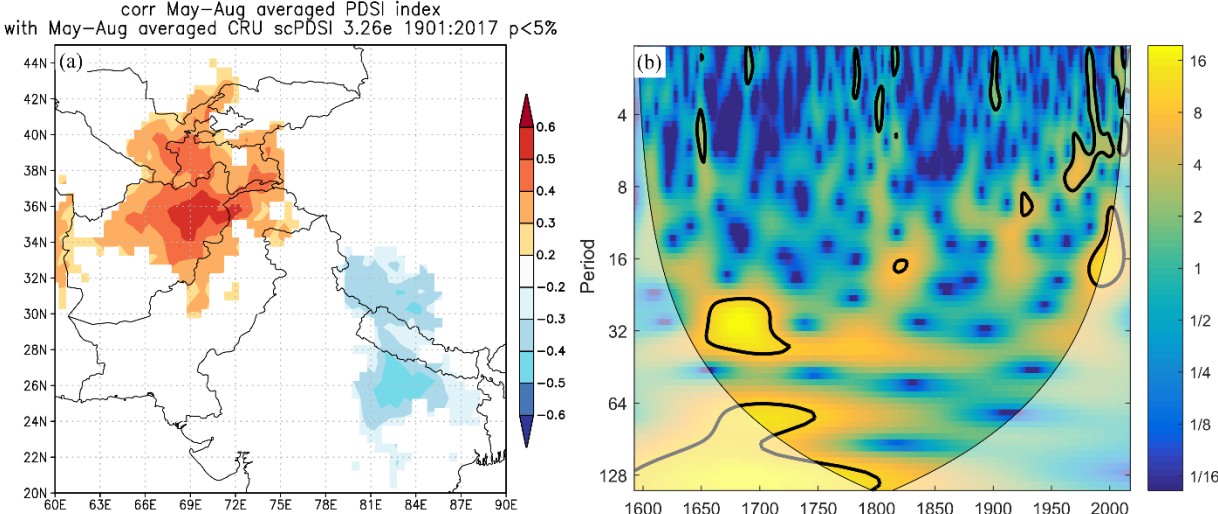


**Fig. 8** (a) Spatial correlation between the actual May-August PDSI and the reconstructed May-
August scPDSI (1901-2017). (b) The wavelet analysis of the reconstructed scPDSI in the Chitral
HinduKush Ranges, Pakistan. The 95% significance level against red noise was shown as a black
contour.







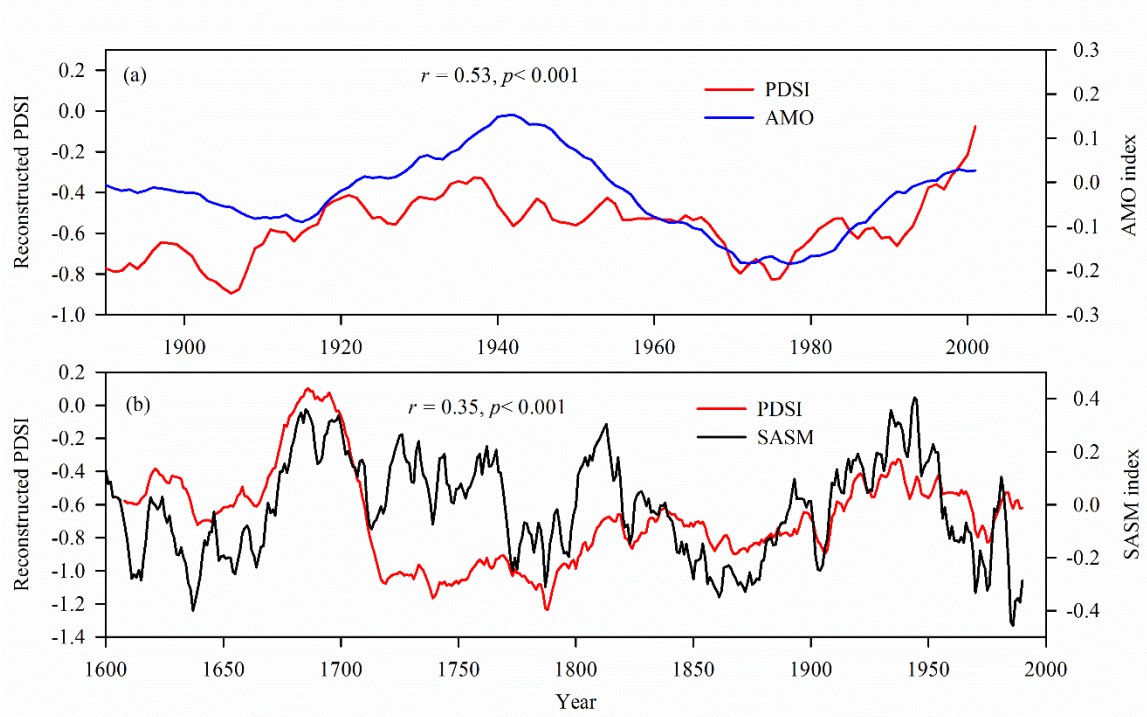

**Fig. 9** (a) Comparison of the 31-year moving average series between the reconstructed Mar-Aug

scPDSI and the AMO index during the common period (1890-2001, Mann et al., ); (b)

Comparison of the 31-year moving average series between the reconstructed Mar-Aug scPDSI

and the South Asian Summer Monsoon index from June to August (JJA-SASM) (Cook et al.,

2010) during the common period (1608-1990).

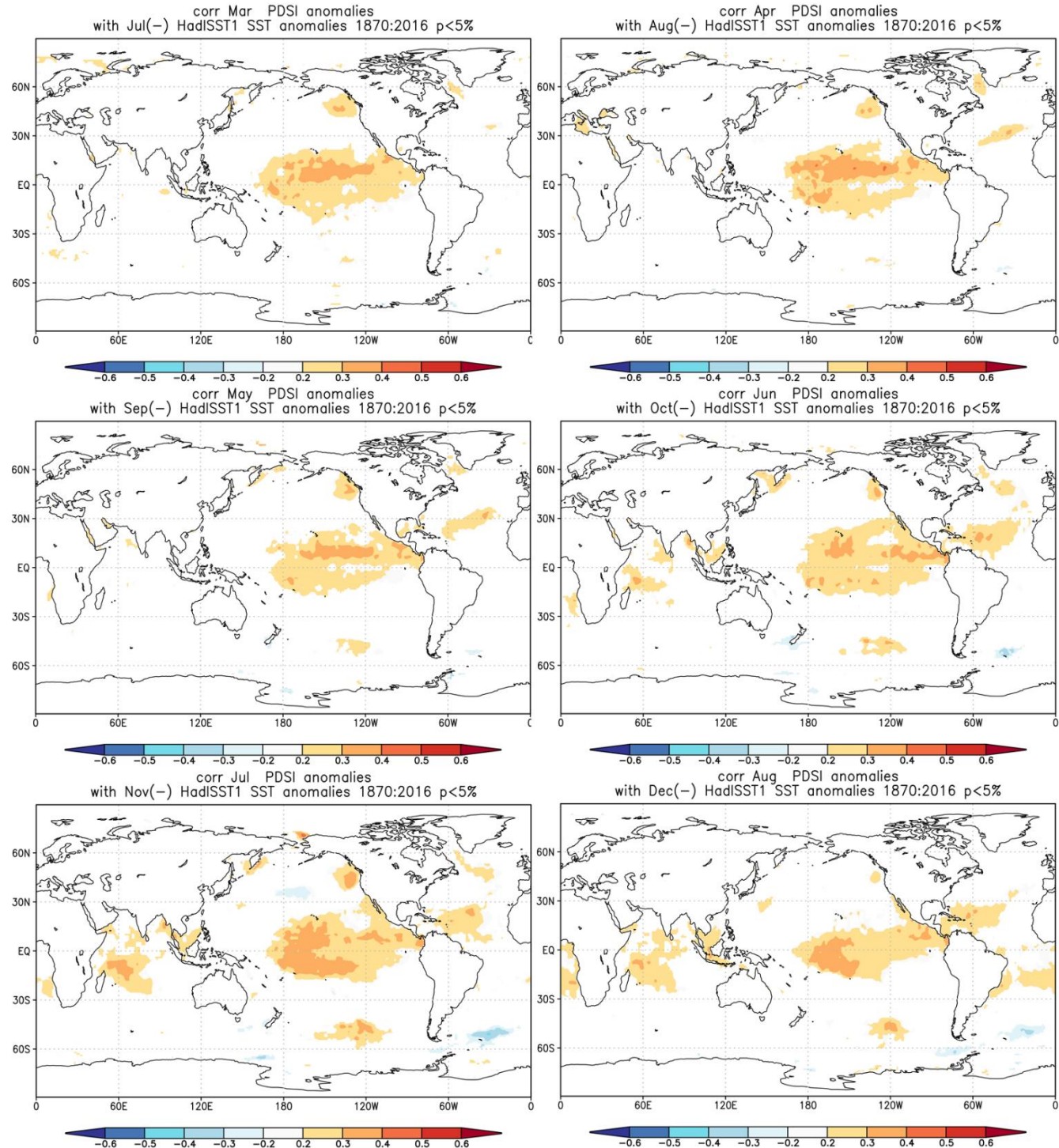


**Fig. 10** The field correlation between the monthly HadISST1 sea surface temperature and reconstructed PDSI with a lag of 8 months calculated by the KNMI Climate Explorer (1870-2016). The contours with $p > 0.05$ were masked out.

**Table 1.** Statistical test for the tree-ring reconstruction of March-August PDSI in Chitral
HinduKush Range of northern Pakistan based on a split calibration-verification procedure.

| Calibrations | $r$ | $R^2$ | Verification | RE | CE | ST | DW | RMSE | PMT |
|---|---|---|---|---|---|---|---|---|---|
| 1960-2016 | 0.70 | 0.49 | — | 0.44 | — | (43, 14)* | 1.06* | 1.21 | 10.0* |
| 1989-2016 | 0.82 | 0.67 | 1960-1988 | 0.61 | 0.62 | (23, 6)* | 1.0* | 1.72 | 5.80* |
| 1960-1988 | 0.73 | 0.53 | 1989-2016 | 0.64 | 0.62 | (24, 4)* | 0.98* | 1.56 | 7.42* |

Notes: RE-Reduction of error, CE-Coefficient of efficiency, ST-Sign test, DW-Durbin-Watson
test, RMSE-Root mean square error, PMT-Product means test.

**Table 2.** Correlation coefficients ($r$) and $p$ value between monthly ENSO index and reconstructed
PDSI with a lag of 8 months calculated by the KNMI Climate Explorer.

| PDSI Month | ENSO Month | NINO3 | | NINO3.4 | | NINO4 | 888 |
|---|---|---|---|---|---|---|---|
| | | $r$ | $p$ | $r$ | $p$ | $r$ | $p$ |
| Jan | May | 0.19 | 0.0445 | 0.21 | 0.0270 | 0.26 | 0.0063 |
| Feb | Jun | 0.23 | 0.0156 | 0.26 | 0.0053 | 0.28 | 0.0028 |
| **Mar** | **Jul** | **0.25** | **0.0094** | **0.28** | **0.0030** | **0.27** | **0.0043** |
| **Apr** | **Aug** | **0.22** | **0.0226** | **0.25** | **0.0083** | **0.26** | **0.0087** |
| **May** | **Sep** | **0.22** | **0.0202** | **0.26** | **0.0074** | **0.28** | **0.0045** |
| **Jun** | **Oct** | **0.18** | **0.0599** | **0.24** | **0.0117** | **0.29** | **0.0033** |
| **Jul** | **Nov** | **0.19** | **0.0488** | **0.25** | **0.0078** | **0.28** | **0.0033** |
| **Aug** | **Dec** | **0.16** | **0.0773** | **0.22** | **0.0157** | **0.26** | **0.0049** |
| Sep | Jan | 0.20 | 0.0432 | 0.24 | 0.0103 | 0.26 | 0.0057 |
| Oct | Feb | 0.26 | 0.0061 | 0.30 | 0.0010 | 0.28 | 0.0031 |
| Nov | Mar | 0.27 | 0.0038 | 0.28 | 0.0020 | 0.27 | 0.0040 |
| Dec | Apr | 0.25 | 0.0090 | 0.27 | 0.0030 | 0.31 | 0.0009 |