# Peer review of "1. Introduction"

_Climate of the Past, 2019_

## Referee Comment (RC1) · Anonymous Referee #1 · 17 Oct 2019

Ahmad et al. contribute a new reconstruction of scPDSI in Northern Pakistan, where such work is still highly necessary. Generally, the manuscript is well organised and the methods are regular and solid, the main conclusions are largely supported by the results. I would be happy to see the work published in CP, but only after some of following concerns well addressed.

Major comments:

1. The compared reference reconstruction Treydte et al. (2006) are generally in opposite phases in the history, but not well explained. It is also controversial to the statement in the abstract (L28-30).

2. More explanations should be provided for the linkage between the climate reconstruction with these ocean oscillation indices in the discussion part.

Specific comments:

L57: What kind of production? Food?

L64: "an essential archive of dendroclimatic research" A lot of tree-ring work done there? Do you really mean this?

L117-119: The drought index described here is not consistent with the results in the Fig. 8a.

L169-170: It is not clear why not comparing with reconstructions in the study area but with that outside of the study region. Please add more explanations.

L181-2: Not clear which criterion was used? EPS or tree No.?

L236: What do you mean by the "point years"? Not mentioned before.

L247-266: I suggest providing the compared reconstructions in the figure if possible.

4.2 The linkage of drought variation with the ocean oscillations: It is a bit stretching to relate the periodicity results to the ocean oscillation indices, more direct proof should be provided, such as direct comparison or, at least, previous work on the actual connections between local climates and there synoptic indices.

Fig 1: This is a very nice figure but the statement on the copyright is quite confusing. Please consider remove or revise it properly.

Fig 2: I suggest provided error bars for the mean values of the climate variables.

Fig 3: I suggest switching the Y-coordinates of the chronology and the sample depth. Besides, I suggest replacing "Tree number" by "The number of trees".

Fig 4: You should explain the meaning of the "previous" and "current" because it is not that self-explaining for every reader. The colors of the two types of the columns are too close, please consider change into other colors.

Fig 6: The location of the Treydte et al. (2006) should be provided in the Fig.1.

Fig 7: The "MTM" should be referred as the full name.

Fig 8: The sentence "The value for p>0.05 were masked out" should be clearly pointed to the Fig. 8a. Again, it is very confusing about the copyright statement.

Table 1: The figure caption should be expanded with more information input to make it be able to stand alone. The stats, such as RE and CE, should be explained in detail here too.

---

## Referee Comment (RC2) · Anonymous Referee #2 · 26 Oct 2019

Ahmad et al. presented a reconstructed PDSI time series from tree-ring record for the HinduKush Range of Pakistan where such record is limited. This study will be valuable for understanding long-term drought dynamics in these regions. The method is typical for this type of research, statistical analyses are sound, and results are checked with existing studies. Overall, I find this study is publishable after addressing the following concerns.

1. The logic flow of introduction section is very unclear. The second paragraph is hard to follow. From line#58-61, it is hard to follow why "their distinct and complex

topography, …….., and uniquie precipitations seasonaility" makes tress in HKH "important"? In addition, what is the purpose of mentioning the hydroelectric reservoir? What is the purpose of comparing 4% from government and 2% from FAO? Moreover, from Line#73-75, why susceptible to ET, soil and air temperature, air humidity, and soil moisture makes tree ring highly recommended for dendroclimatic studies? I highly recommend authors fill in these missing logic links to improve readability. 2. Line#153. Do you have a reason for reconstructing March-August PDSI rather than other period or for the annual mean PDSI? This should be explained in the main text. 3. Line#166-168. Do you have a reason for choosing 1 standard deviation to identify drought/wet periods? Please explain. In addition, the purpose of the second sentence "We assessed the dry and wet periods for many years based on strength and intensity" is not clear to me. 4. In Fig. 5, the 1960-2016 is relatively dry. This could bias your regression equation at Line#198 towards the dry end. I think this is the reason why the mean of your reconstructed PDSI before 1960 is negative rather than zero. How did you correct this dry bias? Please explain and add discussions on how this dry bias would affect your drought identification and conclusions. 5. Figs. 6 and 8 are unreferenced in the main text. Please correct. In addition, please consider add more explanations for these two figures. Currently, it is unclear how these two figures support the flow of your discussions.

The English of this manuscript needs to be polished. The following are a list of errors that I caught. • Line 48, lacking an "and" before "increases risk of wildfires". • Line 57, 80% to 60% of what? • Line 63, replace "in the form of" with "from" • Line 169, remove "a" before "few", remove "still". • Line 185-187, please consider decompose this long sentence into smaller ones. • Line 308, is "weather" a good word for climate-related studies?

---

## Author Comment (AC1) · 11 Dec 2019

Dear Prof. Hans Linderholm and two reviewers,

Thank you very much for giving us a chance to revise our manuscript "A 424-year tree-ring based PDSI reconstruction of Cedrus deodara D. Don from Chitral HinduKush Range of Pakistan: linkages to the ocean oscillations" (cp-2019). We greatly appreciate the two reviewers for their valuable and helpful comments. We have revised our manuscript according their comments. The manuscript has been greatly improved after the revision. The point by point response to the reviewers' comments can be found in the following pages. The reviewer's comments are listed in black, and our response are in blue. Thank you for all your help in processing our manuscript. We look forward to hearing from you soon.

Best wishes,

Xiaochun Wang

On behalf of all co-authors

Corresponding author: Xiaochun Wang at School of Forestry, Northeast Forestry University, Harbin 150040, China,

Phone: +86 451 82190509

E-mail address: wangx@nefu.edu.cn

Anonymous Referee #1:

Ahmad et al. contribute a new reconstruction of scPDSI in Northern Pakistan, where Such work is still highly necessary. Generally, the manuscript is well organized and the methods are regular and solid, the main conclusions are largely supported by the Results. I would be happy to see the work published in CP, but only after some of following concerns well addressed.

Response: Thank you for your affirmation. We have revised it carefully according to your comments.

1. The compared reference reconstruction Treydte et al. (2006) are generally in opposite phases in the history, but not well explained. It is also controversial to the statement in the abstract (L28-30).

Response: Fully accepted. We used the raw $\delta18O$ value in Treydte et al. (2006) to compare with our reconstruction, while the high raw $\delta18O$ value represents dry and

the low value represents wet, just opposite to the PDSI index. Therefore, their results are just the opposite of ours. We have added explained the situation in the manuscript.

2. More explanations should be provided for the linkage between the climate reconstructions with these ocean oscillation indices in the discussion part.

Response: Fully accepted. More explanations have been provided for the linkage between the climate reconstructions with these ocean oscillation indices in the discussion section. Khan et al. (2014) showed that most of our study area is covered by monsoon shadow, but the Asian monsoon showed an overall weak trend in recent decades (Li and Zeng, 2003; Wang and Ding, 2006; Ding et al., 2008). Therefore, the increase of regional precipitation may be linked to the ENSO. Previous studies (Chen et al., 2019; Shi et al., 2106, Wang et al., 2005) have confirmed that ENSO is an important factor in regulating the hydrological conditions related to the AMO. In the past, sever famine and drought occurred simultaneously with the warm phase of ENSO, and these events were related to the failure of Indian Summer Monsoon (Shi et al., 2014b; Shi et al., 2014a).

L57: What kind of production? Food?

Response: Fully accepted. Agricultural production has been added.

The long-term drought from 1998 to 2002 reduced agricultural production, with the largest reduction in wheat, barley and sorghum (from 60% to 80%) (Ahmad et al., 2004).

L64: "an essential archive of dendroclimatic research" A lot of tree-ring work done there? Do you really mean this?

Response: The sentence has been removed.

L117-119: The drought index described here is not consistent with the results in the Fig. 8a.

Response: It is the same PDSI data. The PDSI in the Fig. 8a was also the CRU self-calibrating PDSI.

L169-170: It is not clear why not comparing with reconstructions in the study area but with that outside of the study region. Please add more explanations.

Response: Treydte et al. (2006) are the closest reconstructions to our research site, almost in our research area. We didn't find any more nearby drought (or precipitation) reconstruction to compare.

L181-2: Not clear which criterion was used? EPS or tree No.?

Response: Fully accepted. Corrected.

According to the threshold of EPS (EPS > 0.85), 1593-2016 was selected as the reconstruction period to truncate the period 1537-1593 of the chronology (Fig. 3).

L236: What do you mean by the "point years"? Not mentioned before

Response: It means the narrow years of tree rings and has been confirmed as dry year by previous studies. We added the explanation of point years. The point years (narrow rings), 2002, 2001, 2000, 1999, 1985, 1971, 1962, 1952, 1945, 1921, 1917, 1902 and 1892, were recorded in our tree-ring record. The narrow ring formation occurs when extreme drought stress reduces cell division (Shi et al., 2014b; Fritts et al., 1976). Therefore, the narrow rings are also consistent with the extreme drought years.

L247-266: I suggest providing the compared reconstructions in the figure if possible.

Response: Fully accepted. However, in this region, the reconstruction of temperature, precipitation and PDSI in the past is less, and the existing reconstruction data in other regions are difficult to obtain. Therefore, we didn't draw such a comparative picture.

4.2 The linkage of drought variation with the ocean oscillations: It is a bit stretching to relate the periodicity results to the ocean oscillation indices, more direct proof should be provided, such as direct comparison or, at least, previous work on the actual connections between local climates and these synoptic indices.

Response: Fully accepted. This discussion was added. The spatial correlation exhibited the significant similarity of El Niño-Southern Oscillation (ENSO) in the region (Figure 8). The intensity of India monsoon in this area was modulated by ENSO patterns. The high frequency of drought cycle (2.1-3.3 years) may be related to the ENSO (Van Oldenborgh and Burgers, 2005). Previous researches (Chen et al., 2019; Shi et al., 2106, Wang et al., 2005) further proved that ENSO is the responsible factor for regulating the hydrological conditions in our study area related to AMO. In the past, sever famines and droughts occurred simultaneously with the El Niño (ENSO warm phase), and these events were related to the failure of Indian Summer Monsoon (Shi et al., 2014b; Shi et al., 2014a).

Fig 1: This is a very nice figure but the statement on the copyright is quite confusing. Please consider remove or revise it properly.

Response: Fully accepted. The copyright has been removed.

Fig 2: I suggest provided error bars for the mean values of the climate variables.

Response: Fully accepted. Done.

Fig 3: I suggest switching the Y-coordinates of the chronology and the sample depth. Besides, I suggest replacing "Tree number" by "The number of trees".

Response: Fully accepted. Done.

Fig 4: You should explain the meaning of the "previous" and "current" because it is not that self-explaining for every reader. The colors of the two types of the columns are too close, please consider change into other colors.

Response: Fully accepted. The figure caption was modified. Also, the bar colors were changed. Fig. 4 Pearson correlation coefficients between the tree-ring index of C. deodara and monthly total precipitation (1965-2013) and scPDSI (1960-2013) (a)

[Figure]

and monthly maximum and minimum temperature (1965-2013) (b) from June of the previous year to September of the current year. Significant correlations (p<0.05) are denoted by asterisks.

Fig 6: The location of the Treydte et al. (2006) should be provided in the Fig.1.

Response: Fully accepted. The location of the Treydte et al. (2006) was added in the Fig. 1.

Fig 7: The "MTM" should be referred as the full name.

Response: Fully accepted. The full name was added. Fig. 7 The Multi-taper method spectrums of the reconstructed scPDSI from 1593 to 2016. Red and green line represents the 95% and 99% confidence level, respectively. The figures above the significant line represents the significant periods of drought at 95% confidence level.

Fig 8: The sentence "The value for p>0.05 were masked out" should be clearly pointed to the Fig. 8a. Again, it is very confusing about the copyright statement.

Response: Fully accepted. The figure caption was modified. Fig. 8 (a) Spatial correlation between the actual May-August scPDSI and the reconstructed May-August scPDSI (1901-2017). (b) The wavelet analysis of the reconstructed scPDSI in the Chitral HinduKush Ranges, Pakistan. The 95% significance level against red noise was shown as a black contour.

Table 1: The table caption should be expanded with more information input to make it be able to stand alone. The stats, such as RE and CE, should be explained in detail here too.

Response: Fully accepted. The table caption was modified. The statistica parameters were explained using notes under the table. Notes: RE-Reduction of error, CE-Coefficient of efficiency, ST-Sign test, DW-Durbin-Watson test, RMSE-Root mean square error, PMT-Product means test.

Anonymous Referee #2

Ahmad et al. presented a reconstructed PDSI time series from tree-ring record for the HinduKush Range of Pakistan where such record is limited. This study will be valuable for understanding long-term drought dynamics in these regions. The method is typical for this type of research, statistical analyses are sound, and results are checked with existing studies. Overall, I find this study is publishable after addressing the following concerns.

Response: Thank you for your affirmation of our manuscript. We have revised the manuscript comprehensively according to your comments and suggestions.

The logic flow of introduction section is very unclear.

Response: Fully accepted. We have deleted some unclear paragraph and revised the introduction.

[revised manuscript text omitted]

The second paragraph is hard to follow. From line#58-61, It is hard to follow why "their distinct and complex topography, and unique precipitations seasonality" makes tress in HKH "important"? In addition, what is the purpose of mentioning the hydroelectric reservoir? What is the purpose of comparing 4% from government and 2% from FAO?

Response: Fully accepted. The second paragraph and the whole the introduction section has been revised. Pakistan has a semi-arid climate, and its agriculture economy is most vulnerable to drought (Kazmi et al., 2015; Miyan, 2015). The long-term drought from 1998 to 2002 reduced agricultural production, with the largest reduction in wheat,

barley and sorghum (from 60% to 80%) (Ahmad et al., 2004). The northern Pakistan is considered to be the world's largest area of irrigation network (Treydte et al., 2006). The production and life of local residents are strongly dependent on monsoon precipitation brought by the mighty ocean and atmospheric circulation system, including El-Nino Southern Oscillation (ENSO), Atlantic Multi-decadal Oscillation (AMO), Pacific Decadal Oscillation (PDO) and others (Treydte et al., 2006; Cook et al., 2010; Miyan, 2015; Zhu et al., 2017). However, the current warming rate has changed the regional hydrological conditions, leading to an unsustainable water supply (Hellmann et al., 2016; Wang et al., 2017). It is not only critical for agricultural production but also leads to forest mortality, vegetation loss (Martínez-Vilalta and Lloret, 2016) and increases the risk of wildfires (Turner et al. 2015; Abatzoglou and Williams 2016). The degradation of grassland and loss of livestock caused by drought eventually affect the lifestyle of nomadic peoples, especially in high-altitude forested areas (Pepin et al., 2015; Shi et al., 2019).

Line#73-75, why susceptible to ET, soil and air temperature, air humidity, and soil moisture makes tree ring highly recommended for dendroclimatic studies?

Response: Fully accepted. This sentence has been removed.

2. Line#153. Do you have a reason for reconstructing March-August PDSI rather than other period or for the annual mean PDSI? This should be explained in the main text.

Response: Fully accepted. The period of March-August has been removed, and why reconstructed the period of March-August was explained in the main text. The correlation between and TRI was the highest from March to August, indicating that the growth of C. deodara was most strongly affected by the drought before and during the growing season. Based on the above correlation analysis results, the March-August PDSI was the most suitable for seasonal reconstruction.

3. Line#166-168. Do you have a reason for choosing 1 standard deviation to identify drought/wet periods? Please explain. In addition, the purpose of the second sentence "We assessed the dry and wet periods for many years based on strength and intensity"

is not clear to me.

Response: Fully accepted. According to the reference of Chen et al. (2019), we defined the periods are above or low the mean ±1 standard deviation of PDSI as the wet or drought periods. Chen, F., Zhang, T., Seim, A., Yu, S., Zhang, R., Linderholm, H. W., Kobuliev, Z. V., Ahmadov, A., and Kodirov, A.: Juniper tree-ring data from the Kuramin Range (northern Tajikistan), reveals changing summer drought signals in western Central Asia. Forests, 10(6), 505, 2019.

4. In Fig. 5, the 1960-2016 is relatively dry. This could bias your regression equation at Line#198 towards the dry end. I think this is the reason why the mean of your reconstructed PDSI before 1960 is negative rather than zero. How did you correct this dry bias? Please explain and add discussions on how this dry bias would affect your drought identification and conclusions.

Response: The reconstructed PDSI seems to be drier than the actual PDSI, mainly because the amplitude range of the reconstruction value is always lower than the actual value, so it seems that the reconstructed PDSI is drier than the actual value. That's also why the mean value of our reconstruction is lower than 0. This phenomenon exists in many climate reconstructions.

5. Figs. 6 and 8 are unreferenced in the main text. Please correct. In addition, please consider add more explanations for these two figures. Currently, it is unclear how these two figures support the flow of your discussions.

Response: Fully accepted. The figure has been texted and explained to support the flow of our discussion part. The results were compared with adjacent studies for validation and reliability (Treydte et al., 2006), they used annually resolved oxygen isotope ($\delta$18O) record from tree rings (Fig. 6). The spatial correlation analysis between our reconstructed and actual PDSI from May to August shows that our drought reconstruction is a good regional representative (Fig. 8). This shows that our reconstruction is reliable and can reflect the drought situation in the region. The spatial correlation between

Actual May-August scPDSI (1901-2017) and Actual May-August reconstructed scPDSI (1901-2017) exhibit a significance effect of El Niño-Southern Oscillation (ENSO) exist in the region (Fig. 8).

The English of this manuscript needs to be polished. The following are a list of errors that I caught. âËŸA ′c Line 48, lacking an "and" before "increases risk of wildfires". âËŸA ′c Line 57, 80% to 60% of what? âËŸA ′c Line 63, replace "in the form of" with "from" âËŸA ′c Line 169, remove "a" before "few", remove "still". âËŸA′c Line 185-187, please consider decompose this long sentence into smaller ones. âËŸA ′c Line 308, is "weather" a good word for climate-related studies?

Response: Fully accepted. Done. The English of the whole manuscript was also polished by the AJE English editing service company.

Please also note the supplement to this comment:
https://www.clim-past-discuss.net/cp-2019-62/cp-2019-62-AC1-supplement.pdf

[Figure]

[Figure]

**Fig. 1.** Map of the weather stations (Drosh station) and sampling sites in the Chitral, HinduKush Mountains, Pakistan. Different colors represent the elevation changes of the study area.

[Figure]

**Fig. 2.** Monthly maximum, mean, minimum temperature (oC) and total precipitation (mm) in the Drosh Weather Station (35.07° N, 71.78° E, 1465 m), Pakistan (1965-2013).

[Figure]

**Fig. 3.** The regional tree-ring width chronology from 1550 to 2017 in the Chitral, HinduKush Mountains, Pakistan. The gray area represents the sample depth.

[Figure]

**Fig. 4.** Pearson correlation coefficients between the tree-ring index of C. deodara and monthly total precipitation (1965-2013) and scPDSI (1960-2013) (a) and monthly maximum and minimum temperature (1965-2013

[Figure]

**Fig. 5.** The scPDSI reconstruction in the Chitral HinduKush Mountain, Pakistan. (a) Comparison between the reconstructed (black line) and actual (red line) scPDSI; (b) The variation of annual (black solid line

[Figure]

**Fig. 6.** Comparison of our PDSI reconstruction (a) with the precipitation reconstruction (tree-ring $\delta$18O) of Treydte et al. (2006) (b) in northern Pakistan. Purple and brown shaded areas represent the consiste

Fig. 7. The Multi-taper method spectrums of the reconstructed scPDSI from 1593 to 2016. Red and green line represents the 95% and 99% confidence level, respectively. The figures above the significant line rep

[Figure]

**Fig. 8.** (a) Spatial correlation between the actual May-August PDSI and the reconstructed May-August scPDSI (1901-2017). (b) The wavelet analysis of the reconstructed scPDSI in the Chitral HinduKush Ranges, Pa

---

## Author Response (AR3)

**Dear Prof. Hans Linderholm and two reviewers,**

Thank you very much for giving us a chance to revise our manuscript "**A 424-year tree-ring based PDSI reconstruction of Cedrus deodara D. Don from Chitral HinduKush Range of Pakistan: linkages to the ocean oscillations**" (cp-2019). We greatly appreciate the two reviewers for their valuable and helpful comments. We have revised our manuscript according their comments. The manuscript has been greatly improved after the revision. The point by point response to the reviewers' comments can be found in the following pages. The reviewer's comments are listed in black, and our response are in blue. Thank you for all your help in processing our manuscript. We look forward to hearing from you soon.

Best wishes,

Xiaochun Wang
On behalf of all co-authors

Corresponding author: Xiaochun Wang at School of Forestry, Northeast Forestry University, Harbin 150040, China,
Phone: +86 451 82190509
E-mail address: wangx@nefu.edu.cn

**Anonymous Referee #1:**

Ahmad et al. contribute a new reconstruction of scPDSI in Northern Pakistan, where Such work is still highly necessary. Generally, the manuscript is well organized and the methods are regular and solid, the main conclusions are largely supported by the Results. I would be happy to see the work published in CP, but only after some of following concerns well addressed.

**Response:** Thank you for your affirmation. We have revised it carefully according to your comments.

1. The compared reference reconstruction Treydte et al. (2006) are generally in opposite phases in the history, but not well explained. It is also controversial to the statement in the abstract (L28-30).

**Response:** Fully accepted. We used the raw $\delta^{18}O$ value in Treydte et al. (2006) to compare with our reconstruction, while the high raw $\delta^{18}O$ value represents dry and the low value represents wet, just opposite to the PDSI index. Therefore, their results are just the opposite of ours. We have added explained the situation in the manuscript.

2. More explanations should be provided for the linkage between the climate reconstructions with these ocean oscillation indices in the discussion part.

**Response:** Fully accepted. More explanations have been provided for the linkage between the climate reconstructions with these ocean oscillation indices in the discussion section.

[revised manuscript text omitted]

L64: "an essential archive of dendroclimatic research" A lot of tree-ring work done there? Do you really mean this?

**Response:** The sentence has been removed.

L117-119: The drought index described here is not consistent with the results in the Fig. 8a.

**Response:** It is the same PDSI data. The PDSI in the Fig. 8a was also the CRU self-calibrating PDSI.

L169-170: It is not clear why not comparing with reconstructions in the study area but with that outside of the study region. Please add more explanations.

**Response:** Treydte et al. (2006) are the closest reconstructions to our research site, almost in our research area. We didn't find any more nearby drought (or precipitation) reconstruction to compare.

L181-2: Not clear which criterion was used? EPS or tree No.?

**Response:** Fully accepted. Corrected.

According to the threshold of EPS (EPS > 0.85), 1593-2016 was selected as the reconstruction period to truncate the period 1537-1593 of the chronology (Fig. 3).

L236: What do you mean by the "point years"? Not mentioned before

**Response:** It means the narrow years of tree rings and has been confirmed as dry year by previous studies. We added the explanation of point years. The point years (narrow rings), 2002, 2001, 2000, 1999, 1985, 1971, 1962, 1952, 1945, 1921, 1917, 1902 and 1892, were recorded in our tree-ring record. The narrow ring formation occurs when extreme drought stress reduces cell division (Shi et al., 2014b; Fritts et al., 1976). Therefore, the narrow rings are also consistent with the extreme drought years.

L247-266: I suggest providing the compared reconstructions in the figure if possible.

**Response:** Fully accepted. However, in this region, the reconstruction of temperature, precipitation and PDSI in the past is less, and the existing reconstruction data in other regions are difficult to obtain. Therefore, we didn't draw such a comparative picture.

4.2 The linkage of drought variation with the ocean oscillations: It is a bit stretching to relate the periodicity results to the ocean oscillation indices, more direct proof should be provided, such as direct comparison or, at least, previous work on the actual connections between local climates and these synoptic indices.

**Response:** Fully accepted. This discussion was added. The spatial correlation exhibited the significant similarity of El Niño-Southern Oscillation (ENSO) in the region (Figure 8). The intensity of India monsoon in this area was modulated by ENSO patterns. The high frequency of drought cycle (2.1-3.3 years) may be related to the ENSO (Van Oldenborgh and Burgers, 2005). Previous researches (Chen et al., 2019; Shi et al., 2106, Wang et al., 2005) further proved that ENSO is the responsible factor for regulating the hydrological conditions in our study area related to AMO. In the past, sever famines and droughts occurred simultaneously with the El Niño (ENSO warm phase), and these events were related to the failure of Indian Summer Monsoon (Shi et al., 2014b; Shi et al., 2014a).

Fig 1: This is a very nice figure but the statement on the copyright is quite confusing.

Please consider remove or revise it properly.

**Response:** Fully accepted. The copyright has been removed.

Fig 2: I suggest provided error bars for the mean values of the climate variables.

**Response:** Fully accepted. Done.

[Figure]

Fig 3: I suggest switching the Y-coordinates of the chronology and the sample depth.
Besides, I suggest replacing "Tree number" by "The number of trees".

**Response:** Fully accepted. Done.

[Figure]

Fig 4: You should explain the meaning of the "previous" and "current" because it is not that self-explaining for every reader. The colors of the two types of the columns are too close, please consider change into other colors.

**Response:** Fully accepted. The figure caption was modified. Also, the bar colors were changed.

Fig. 4 Pearson correlation coefficients between the tree-ring index of *C. deodara* and monthly total precipitation (1965-2013) and scPDSI (1960-2013) (a) and monthly maximum and minimum temperature (1965-2013) (b) from June of the previous year to September of the current year. Significant correlations ($p<0.05$) are denoted by asterisks.

[Figure]

Fig 6: The location of the Treydte et al. (2006) should be provided in the Fig.1.

**Response:** Fully accepted. The location of the Treydte et al. (2006) was added in the Fig. 1.

Fig 7: The "MTM" should be referred as the full name.

**Response:** Fully accepted. The full name was added.

Fig. 7 The Multi-taper method spectrums of the reconstructed scPDSI from 1593 to 2016. Red and green line represents the 95% and 99% confidence level, respectively. The figures above the significant line represents the significant periods of drought at 95% confidence level.

Fig 8: The sentence "The value for $p>0.05$ were masked out" should be clearly pointed to the Fig. 8a. Again, it is very confusing about the copyright statement.

**Response:** Fully accepted. The figure caption was modified.

Fig. 8 (a) Spatial correlation between the actual May-August scPDSI and the reconstructed May-August scPDSI (1901-2017). (b) The wavelet analysis of the reconstructed scPDSI in the Chitral HinduKush Ranges, Pakistan. The 95% significance level against red noise was shown as a black contour.

Table 1: The table caption should be expanded with more information input to make it be able to stand alone. The stats, such as RE and CE, should be explained in detail here too.

**Response:** Fully accepted. The table caption was modified. The statistica parameters were explained using notes under the table.

**Table 1.** Statistical test for the tree-ring reconstruction of March-August PDSI in Chitral HinduKush Range of northern Pakistan based on a split calibration-verification procedure.

| Calibrations | $r$ | $R^2$ | Verification | RE | CE | ST | DW | RMSE | PMT |
|---|---|---|---|---|---|---|---|---|---|
| 1960-2016 | 0.70 | 0.49 | — | 0.49 | — | (43, 14)* | 1.06* | 1.21 | 10.0* |
| 1989-2016 | 0.82 | 0.67 | 1960-1988 | 0.61 | 0.62 | (23, 6)* | 1.0* | 1.72 | 5.80* |
| 1960-1988 | 0.73 | 0.53 | 1989-2016 | 0.64 | 0.62 | (24, 4)* | 0.98* | 1.56 | 7.42* |

Notes: RE-Reduction of error, CE-Coefficient of efficiency, ST-Sign test, DW-Durbin-Watson test, RMSE-Root mean square error, PMT-Product means test.

**Anonymous Referee #2**

Ahmad et al. presented a reconstructed PDSI time series from tree-ring record for the HinduKush Range of Pakistan where such record is limited. This study will be valuable for understanding long-term drought dynamics in these regions. The method is typical for this type of research, statistical analyses are sound, and results are checked with existing studies. Overall, I find this study is publishable after addressing the following concerns.

**Response:** Thank you for your affirmation of our manuscript. We have revised the manuscript comprehensively according to your comments and suggestions.

The logic flow of introduction section is very unclear.

**Response:** Fully accepted. We have deleted some unclear paragraph and revised the introduction.

[revised manuscript text omitted]

The second paragraph is hard to follow. From line#58-61, It is hard to follow why "their distinct and complex topography, and unique precipitations seasonality" makes tress in HKH "important"? In addition, what is the purpose of mentioning the hydroelectric reservoir? What is the purpose of comparing 4% from government and 2% from FAO?

**Response:** Fully accepted. The second paragraph and the whole the introduction section has been revised.

Pakistan has a semi-arid climate, and its agriculture economy is most vulnerable to drought (Kazmi et al., 2015; Miyan, 2015). The long-term drought from 1998 to 2002 reduced agricultural production, with the largest reduction in wheat, barley and sorghum (from 60% to 80%) (Ahmad et al., 2004). The northern Pakistan is considered to be the world's largest area of irrigation network (Treydte et al., 2006). The production and life of local residents are strongly dependent

on monsoon precipitation brought by the mighty ocean and atmospheric circulation system, including El-Nino Southern Oscillation (ENSO), Atlantic Multi-decadal Oscillation (AMO), Pacific Decadal Oscillation (PDO) and others (Treydte et al., 2006; Cook et al., 2010; Miyan, 2015; Zhu et al., 2017). However, the current warming rate has changed the regional hydrological conditions, leading to an unsustainable water supply (Hellmann et al., 2016; Wang et al., 2017). It is not only critical for agricultural production but also leads to forest mortality, vegetation loss (Martínez -Vilalta and Lloret, 2016) and increases the risk of wildfires (Turner et al. 2015; Abatzoglou and Williams 2016). The degradation of grassland and loss of livestock caused by drought eventually affect the lifestyle of nomadic peoples, especially in high-altitude forested areas (Pepin et al., 2015; Shi et al., 2019).

Line#73-75, why susceptible to ET, soil and air temperature, air humidity, and soil moisture makes tree ring highly recommended for dendroclimatic studies?

**Response:** Fully accepted. This sentence has been removed.

2. Line#153. Do you have a reason for reconstructing March-August PDSI rather than other period or for the annual mean PDSI? This should be explained in the main text.

**Response:** Fully accepted. The period of March-August has been removed, and why reconstructed the period of March-August was explained in the main text.

The correlation between and TRI was the highest from March to August, indicating that the growth of *C. deodara* was most strongly affected by the drought before and during the growing season. Based on the above correlation analysis results, the March-August PDSI was the most suitable for seasonal reconstruction.

3. Line#166-168.

Do you have a reason for choosing 1 standard deviation to identify drought/wet periods? Please explain. In addition, the purpose of the second sentence "We assessed the dry and wet periods for many years based on strength and intensity" is not clear to me.

**Response:** Fully accepted. According to the reference of Chen et al. (2019), we defined the periods are above or low the mean ±1 standard deviation of PDSI as the wet or drought periods.

Chen, F., Zhang, T., Seim, A., Yu, S., Zhang, R., Linderholm, H. W., Kobuliev, Z. V., Ahmadov, A., and Kodirov, A.: Juniper tree-ring data from the Kuramin Range (northern Tajikistan), reveals changing summer drought signals in western Central Asia. Forests, 10(6), 505, 2019.

4. In Fig. 5, the 1960-2016 is relatively dry. This could bias your regression equation at Line#198 towards the dry end. I think this is the reason why the mean of your reconstructed PDSI before 1960 is negative rather than zero. How did you correct this dry bias? Please explain and add discussions on how this dry bias would affect your drought identification and conclusions.

**Response: Fully accepted.** The explanation was added in the discussion section.

In Fig. 5, the mean of our reconstructed PDSI is below zero. There are two possible reasons for this phenomenon. First, tree growth is more sensitive to drying than to wetting. As a result, more drought information is recorded in ring widths. This leads to a drier (less than zero) PDSI reconstructed with tree rings. This phenomenon exists in many tree-ring PDSI reconstructions (Hartl-Meier et al., 2017; Wang et al., 2008). Second, the period (1960-2016) used to reconstruct the equation is relatively dry. This cause the mean of the reconstruction equation to be lower than zero (dry), resulting in lower values for the whole reconstructions. Therefore, when applying the PDSI data reconstructed by tree rings, its relative value is relatively reliable, and the absolute value data can only be used after adjustment. The adjustment method of the absolute value needs to be further studied.

5. Figs. 6 and 8 are unreferenced in the main text. Please correct. In addition, please consider add more explanations for these two figures. Currently, it is unclear how these two figures support the flow of your discussions.

**Response:** Fully accepted. The figure has been texted and explained to support the flow of our discussion part. The results were compared with adjacent studies for validation and reliability (Treydte et al., 2006), they used annually resolved oxygen isotope ($\delta^{18}O$) record from tree rings (Fig. 6). The spatial correlation analysis between our reconstructed and actual PDSI from May to August shows that our drought reconstruction is a good regional representative (Fig. 8). This shows that our reconstruction is reliable and can reflect the drought situation in the region.

The spatial correlation between Actual May-August scPDSI (1901-2017) and Actual May- August reconstructed scPDSI (1901-2017) exhibit a significance effect of El Niño-Southern Oscillation (ENSO) exist in the region (Fig. 8).

The English of this manuscript needs to be polished. The following are a list of errors that I caught. ă˘A ´c Line 48, lacking an "and" before "increases risk of wildfires". ă˘A ´c Line 57, 80% to 60% of what? ă˘A ´c Line 63, replace "in the form of" with "from" ă˘A ´c Line 169, remove "a" before "few", remove "still". ă˘A´c Line 185-187, please consider decompose this long sentence into smaller ones. ă˘A ´c Line 308, is "weather" a good word for climate-related studies?

**Response:** Fully accepted. Done. The English of the whole manuscript was also polished by the AJE English editing service company.

**Dear Prof. Hans Linderholm,**

We really appreciate your comments and suggestion on our paper "**A 424-year tree-ring based PDSI reconstruction of *Cedrus deodara* D. Don from Chitral HinduKush Range of Pakistan: linkages to the ocean oscillations**" (cp-2019). We have revised our manuscript according to your comments. In following page, you can find the response to your questions. Your comments are listed in black, and our response are in blue. Thank you for all your help in processing our manuscript. We are looking forward to hearing from you soon.

Best wishes,

Xiaochun Wang

On behalf of all co-authors

Corresponding author: Xiaochun Wang at School of Forestry, Northeast Forestry University,

Harbin 150040, China,

Phone: +86 451 82190509

E-mail address: wangx@nefu.edu.cn

Recommendations:

The language needs to be thoroughly revised. I have started making some changes in the attached pdf, but I don't have the time to check the entire manuscript, and this must be done throughout. Maybe you should contact a language editing service with scientific competence?

**Response: Fully accepted.** The language has been edited by the Elsevier Language Editing Services. The editing certificate was attached.

Line 53 ff.: You should briefly describe the association between precipitation and large-scale atmospheric (ocean) circulation patterns in northern Pakistan and you need references using observations or climate models rather than proxy data to substantiate the claim.

**Response: Fully accepted**. The South Asian summer monsoon (SASM) is an integral component of the global climate system (Cook et al., 2010). Owing to the annually recurring nature of the SASM, it is a significant source of moisture to the subcontinent and to surrounding areas such as northern Pakistan (Betzler et al., 2016). The active phase of the monsoon includes extreme precipitation in the form of floods and heavy snowfall, while the break phase mostly appears in

the form of drought, thereby creating water scarcity. The active/break phases of the monsoon are also concurrent with El Niño-Southern Oscillation (ENSO) and land-sea thermal contrast (Xu et al., 2018; Sinha et al., 2007, 2011). The large-scale variability in sea surface temperature (SST) is induced in the form of Atlantic Multidecadal Oscillation (AMO), Pacific Decadal Oscillation (PDO), and some external forcing, i.e., volcanic eruption and greenhouse gases (Malik et al., 2017; Wei and Lohmann, 2012; Goodman et al., 2005).

Line 78 ff.: You need to provide some more information on dendroclimatological research in Pakistan as I'm sure that more has been done than Treydte et al. 2006 and Khan et al. 2019 (which is not in the reference list).

**Response: Fully accepted.** Before 2010, there are few tree-ring studies in Pakistan. Bilham et al. (1983) found that tree rings of *Juniper* trees from the Sir Sar Range in the Karakoram have the potential to reconstruct past climate. Esper et al. (1995) developed a 1000-year tree-ring chronology at the timberline of Karakorum and found that temperature and rainfall are both controlling factors of *Juniper* growth. More Juniper tree-ring chronologies were developed at the upper timberline in the Karakorum (Esper, 2000; Esper et al., 2001; Esper et al., 2002). *Abies pindrow* and *Picea smithiana* were also used for dendroclimatic investigation in Pakistan (Ahmed et al., 2009; Ahmed et al, 2010). Recently, more studies on tree-rings research have been carried out in Pakistan (Ahmed et al., 2010; Ahmed et al., 2011; Khan et al., 2013; Akbar et al., 2014; Asad et al., 2017a; 2014; Asad et al., 2017b; Asad et al., 2018; Shad et al., 2019), but few have used tree rings to reconstruct the past climate, especially the drought index.

Line 87: How can tree-rings be used to forecasting future climate?

**Response:** It is very few, but do have. Modeling the relationship between tree rings and climate to predict future climate change. A reference (Liu et al., 2004) was added.

Liu, Y., Shishov, V., Shi, J., Vaganov, E., Sun, J., Cai, Q., Djanseitov, I., and An, Z.: The forecast of seasonal precipitation trend at the north Helan Mountain and Baiyinaobao regions, Inner Mongolia for the next 20 years, Chinese Science Bulletin, 49, 410-415, 2004.

Line 108: The sampling site needs to be better described: elevation, aspect, stand density, ground vegetation etc.

**Response: Fully accepted.** The elevation of the study area ranges from 1070 to 7708 m, with an average elevation of 3500 m. The sampled Jigja site is located in the east slope of the mountain.

The stand density is relatively uniform with the dominant species. Among the tree species, *Cedrus deodara* is the most abundant, with 156 individuals' hm$^{-2}$ and basal area of 27 m$^2$ hm$^{-2}$. The Chitral forest is mainly composed of *C. deodara, Juglans regia, Juniperus excelsa, Quercus incana, Quercus dilatata, Quercus baloot*, and *Pinus wallichiana. C. deodara* was selected for sampling because of its high dendroclimatic value (Khan et al., 2013). The soil at our sampling sites was acidic, with little variation within a stand of forest. Similarly, the soil water holding capacity ranged from 47%±2.4% to 62%±4.6% while the soil moisture ranged from 28%±0.57% to 57%±0.49% (Khan et al., 2010).

Line 140: What do you mean by "All false one has been modified…"? Modified in what way? How many false rings were encountered (i.e. were they usual?)?

**Response:** The false one means the tree-ring series with error prompt in the test of COFECHA program. We checked them and revised the errors. So, the false one was not the actual false ring. There are do some false rings, but they are normal, mostly in the latewood.

Line 187: Can you confirm that the EPS is calculated on trees rather than cores? Looking at the tree-ring width chronology, the variance changes considerably back in time. How can this be explained?

**Response:** Yes, we confirm that the EPS is calculated on tree. However, the number of trees is the same as that of the cores because we took one core per tree. Yes, if directly seen from the figure, it seems that the variance decreases back in time. This may be due to the increase of abnormal dry and wet years after 1900 (Duan et al., 2020), similar phenomena also appeared in the series of Treydte et al. (2006).

Duan, J., Wu, P., Ma, Z., and Duan, Y.: Unprecedented recent late-summer warm extremes recorded in tree-ring density on Tibetan Plateau, Environmental Research Letters, 15, 024006.

Line 213-216: Move to method part, not a result.

**Response: Fully accepted.** It has been moved to method part.

Line 263: I assume that you mean pointer years, and these actually include both the most narrow and wide rings. Thus, it would be good to show and discuss also very wet years.

**Response: Fully accepted.** Similarly, the seventeen wettest years found that has been observed from wide rings in the year of 2010, 2009, 2007, 1998, 1997, 1996, 1993, 1931, 1924, 1923, 1908,

1696, 1693, 1691, 1690, 1689, and 1688.

The wet years of 1997, 1996, 1993, 1696, 1693, 1691, 1690, 1689 and 1688 are in agreements with the results of Khan et al., (2019). Similarly the wet years of 1923, 1924, 1988, 2007, 2009 and 2010 coincide with results of Chen et al., (2019) reconstruction.

Line 276: Good discussion, but It almost seems like there are two different chronologies that are combined, with large variance from 1900 until now, and much less before that. I think you need to consider this, and provide as much information on the tree-ring data as possible (see above), including if the trees were all living, if they were sampled at different elevations/environments etc. Does this agree with other drought reconstructions from central Eurasia?

**Response:** We compared this reconstruction with tree-ring-based reconstructions of drought and precipitation from central Eurasia and China, which were adjacent to the northern areas, to test coherency for drought periods, but none of them matched perfectly. The dry periods of our reconstruction showed resemblance with certain periods of 1629–1645 and 1919–1933 of Sun and Liu (2019) reconstruction, while we found more consistent drought periods with He et al.'s (2018) May–June reconstruction from the south-central Tibetan Plateau for 1593–1598 (1580–1598), 1647–1660 (1650–1691), 1785–1800 (1782–1807), and 1870–1878 (1867–1982). The discrepancy might have been caused by the differences in precipitation, geography, species, and reconstruction indexes, among other reasons (Gaire et al., 2019).

It would be informative to compare the reconstructions mentioned in line 286 ff. to yours, as well as indicating the historical droughts mentioned in the text in a figure. I'm pretty sure that you would be able to get hold of those reconstructions.

**Response: Fully accepted.** Our reconstruction featured nine dry and eight wet periods of 1593–1598, 1602–1608, 1631–1645, 1647–1660, 1756–1765, 1785–1800, 1870–1878, 1917–1923, and 1981–1995 and 1663–1675, 1687–1708, 1771–1773, 1806–1814, 1844–1852, 1932–1935, 1965–1969, and 1990–1999, respectively. The dry periods of 1598–1612, 1638–1654, 1753–1761, 1777–1793, and 1960–1985 and the wet periods of 1655–1672, 1681–1696, 1933–1959, and 1762–1776 coincided with that reconstructed by Chen et al. (2019) in northern Tajikistan. The most serious drought in 1871, 1881, and 1931, and the short-term drought from 2000 to 2002 mentioned by Ahmed et al. (2004) were also found to be very dry in our reconstruction.

It would make sense to have all discussion related to the drivers of droughts separated from the comparison between reconstructions. Thus, I suggest moving the sentence in line 293 as well as

the short mention of volcanic influences (which could be expanded)

**Response: Fully accepted.** The sentence in line 293 and the short mention of volcanic influences were removed.

The paragraph regarding the comparison with the Treydte data is a bit confusing, both stating that there is a "strong consistency" and then providing several reasons for why the two records disagree. Maybe it will become a bit clearer if you turn either of the records as suggested above.

Response: **Fully accepted.** Here, the lack of discrepancies means that discrepancies exist with some periods, while in this sentence "In addition, the lack of consistency between different data sets or regions may be due to the dominance of internal climate variability over the impact of natural exogenous forcing conditions on multi-decadal timescales (Bothe et al., 2019)", describing the lack of consistency with the overall differences with other reconstructions. This sentence has been moved to the end of paragraph now.

The discussion regarding the influence of ENSO is very confusing. What do you mean by "The spatial correlation exhibited the significant similarity of El Niño-Southern Oscillation (ENSO) in the region"? The whole section is very speculative, and why not compare the reconstruction with an ENSO record (or at least look into the possible association between droughts/pluvials and El Niño/La Niña years etc.?). Also, while utilising climate explorer, you could see if there are any correlations between drought in N Pakistan and SST in the e.g. Niño 4 region (removing the trends first).There indeed seems to be some connection with AMO, but it would be interesting to see if there are any connections to more close oceans, like the Indian Ocean or the Pacific (which you already hit on regarding ENSO).

**Response: Fully accepted.** The sentence "The spatial correlation exhibited the significant similarity of El Niño-Southern Oscillation (ENSO) in the region" was removed. The correlations between the reconstruction and the ENSO index series (Table 2) and SST field (Fig. 10) were added, and this discussion was revised.

The high frequency of the drought cycle (2.1–3.3 y) may be related to ENSO (van Oldenborgh and Burgers, 2005). The ENSO index in different equator Pacific regions has a significant positive correlation with our reconstructed drought index with a lag of 8 months (Table 2 and Fig. 10), so it further indicated that the water availability in this area may be related to large-scale climate oscillations. There is a lag effect of ENSO on drought in the study area, the lag time is about 4-11 months. The lags in the ENSO impact are very complex and different in different regions (Vicente-Serrano et al., 2011). Therefore, the decrease of drought in our study area may

be linked to the enhancement of ENSO activity. However, Khan et al. (2014) showed that most of northern Pakistan is in the monsoon shadow zone, and the Asian monsoon showed an overall weak trend in recent decades (Wang and Ding, 2006; Ding et al., 2008). Previous studies (Wang et al., 2006; Palmer et al., 2015; Shi et al., 2018; Chen et al., 2019) have confirmed that ENSO is an important factor regulating the hydrological conditions related to the AMO. In the past, severe famine and drought occurred simultaneously with the warm phase of ENSO, and these events were related to the failure of the Indian summer monsoon (Shi et al., 2014).

**Table 2.** Correlation coefficients ($r$) and $p$ value between monthly ENSO index and reconstructed PDSI with a lag of 8 months calculated by the KNMI Climate Explorer.

| PDSI Month | ENSO Month | NINO3 | | NINO3.4 | | NINO4 | |
|---|---|---|---|---|---|---|---|
| | | $r$ | $p$ | $r$ | $p$ | $r$ | $p$ |
| Jan | May | 0.19 | 0.0445 | 0.21 | 0.0270 | 0.26 | 0.0063 |
| Feb | Jun | 0.23 | 0.0156 | 0.26 | 0.0053 | 0.28 | 0.0028 |
| **Mar** | **Jul** | **0.25** | **0.0094** | **0.28** | **0.0030** | **0.27** | **0.0043** |
| **Apr** | **Aug** | **0.22** | **0.0226** | **0.25** | **0.0083** | **0.26** | **0.0087** |
| **May** | **Sep** | **0.22** | **0.0202** | **0.26** | **0.0074** | **0.28** | **0.0045** |
| **Jun** | **Oct** | **0.18** | **0.0599** | **0.24** | **0.0117** | **0.29** | **0.0033** |
| **Jul** | **Nov** | **0.19** | **0.0488** | **0.25** | **0.0078** | **0.28** | **0.0033** |
| **Aug** | **Dec** | **0.16** | **0.0773** | **0.22** | **0.0157** | **0.26** | **0.0049** |
| Sep | Jan | 0.20 | 0.0432 | 0.24 | 0.0103 | 0.26 | 0.0057 |
| Oct | Feb | 0.26 | 0.0061 | 0.30 | 0.0010 | 0.28 | 0.0031 |
| Nov | Mar | 0.27 | 0.0038 | 0.28 | 0.0020 | 0.27 | 0.0040 |
| Dec | Apr | 0.25 | 0.0090 | 0.27 | 0.0030 | 0.31 | 0.0009 |

[Figure]

**Fig. 10** The field correlation between the monthly HadISST1 sea surface temperature and reconstructed PDSI with a lag of 8 months calculated by the KNMI Climate Explorer (1870-2016). The contours with $p > 0.05$ were masked out.

I don't understand the statement on Line 366: "Due to reconstruction indices, species, geographical differences and other reasons, it does not remain the same with the whole period."?

**Response:** This sentence has changed and made it understandable. To test the consistency of the

drought period, we compared this reconstruction with other drought and precipitation based on tree-ring- reconstructions in central Eurasia and China, which were adjacent to our study area, but none of them are completely matched. The dry periods of our reconstruction are similar to some periods of the reconstruction by Sun and Liu (2019) in 1629–1645 and 1919–1933. However, we found that our drought periods are more consistent with the drought periods of May–June reconstruction in the south-central Tibetan Plateau (He et al., 2018), such as 1593–1598 (1580–1598), 1647–1660 (1650–1691), 1785–1800 (1782–1807), and 1870–1878 (1867–1982). This difference may be due to differences in geographical location, species, and reconstruction indices, among others (Gaire et al., 2019).

Line 375: I would be very surprised if there is a consistent EPS>0.85 for only 5 trees throughout the chronology. This is not even achieved using MXD from extremely temperature sensitive trees from high latitudes.
**Response: Full accepted.** The value fore EPS >0.85 start in 1693 (13 trees).

Figure 3. It would be beneficial to include the EPS values in the figure
**Response: Fully accepted.** The EPS value has been added.

[Figure]

Figure 4. This figure is not very nice. Please increase the scale on the left figure (the * are outside the box) and refrain from using red/green colours in the right hand figure (difficult to read for colour blind persons). You also need to include information on which data you used and sources.
**Response: Fully accepted.** Done.

[Figure]

**Fig. 4** Pearson correlation coefficients between the tree-ring index of *C. deodara* and monthly total precipitation (1965-2013) and scPDSI (1960-2013) (a) and monthly maximum and minimum temperature (1965-2013) (b) from June of the previous year to September of the current year. Significant correlations ($p<0.05$) are denoted by asterisks. The "previous" and "current" represents the previous and current year, respectively. The data of monthly precipitation, maximum temperature and minimum temperature were obtained from the meteorological station of Chitral in northern Pakistan. The PDSI data was download from data sets of the grid point (35.36 °N, 71.48 °E) through the Climatic Research Unit (CRU TS.3.22; 0.5° latitude × 0.5° longitude)

Figure 6. Why not reverse the Treydte data to make the two records more comparable?
**Response: Fully accepted.** Done.

[Figure]

**Fig. 6** Comparison of our PDSI reconstruction (a) with the reversed precipitation reconstruction

(tree-ring $\delta^{18}O$) of Treydte et al. (2006) (b) in northern Pakistan. Purple and brown shaded areas represent the consistent wet and dry periods in the two reconstructions, respectively. Two correlation coefficients ($r = 0.24$ and $r = 0.11$) are the correlation of two original annual resolution reconstruction series and two 11-year moving average series, respectively.

Figure 9. Again, you need to cite the data you are using. In the AMO comparison, I suggest removing the trend in your reconstruction (this is done for the AMO). What is SAMS? Why compare with JJA-SAMS?

**Response: Fully accepted.** The data used for AMO reconstruction has been cited "Mann et al., 2009".

The SAMS is the South Asian Summer Monsson. The SASM is one of the important sources of moisture in northern Pakistan (Betzler et al., 2016). June to August is the driest season in northern Pakistan. Therefore, we compared the reconstructed PDSI with the SASM from June to August (JJA-SASM). The JJA-SASM is the available reconstruction for monsoon region downloaded from MADA (Cook et al., 2010).

There is no reference to figure 9 in the text.
**Response: Done. Thanks.**

This showed that our reconstruction was reliable and could reflect the drought situation in the region. In addition, the PDSI of low-frequency (the 31-year moving average) reconstruction had good consistency with the AMO ($r = 0.53$; $p < 0.001$; 1890–2001) and SASM ($r = 0.35$; $p < 0.001$; 1608–1990), which indicated that these are the potential factors affecting the drought patterns in the region (Fig. 9).